# Cryo-EM structures reveal the activation and substrate recognition mechanism of human enteropeptidase

Xiaoli Yang[1,3], Zhanyu Ding [1,2,3], Lisi Peng [1,3], Qiuyue Song [1], Deyu Zhang [1], Fang Cui [1], Chuanchao Xia [1], Keliang Li [1], Hua Yin [1], Shiyu Li [1], Zhaoshen Li [1] & Haojie Huang [1]

Enteropeptidase (EP) initiates intestinal digestion by proteolytically processing trypsinogen, generating catalytically active trypsin. EP dysfunction causes a series of pancreatic diseases including acute necrotizing pancreatitis. However, the molecular mechanisms of EP activation and substrate recognition remain elusive, due to the lack of structural information on the EP heavy chain. Here, we report cryo-EM structures of *human* EP in inactive, active, and substrate-bound states at resolutions from 2.7 to 4.9 Å. The EP heavy chain was observed to clamp the light chain with CUB2 domain for substrate recognition. The EP light chain N-terminus induced a rearrangement of surface-loops from inactive to active conformations, resulting in activated EP. The heavy chain then served as a hinge for light-chain conformational changes to recruit and subsequently cleave substrate. Our study provides structural insights into rearrangements of EP surface-loops and heavy chain dynamics in the EP catalytic cycle, advancing our understanding of EP-associated pancreatitis.

*Human* enteropeptidase (*h*EP), also known as enterokinase, is an essential enzyme in food digestion and is localized at the brush border of the duodenal and jejunal mucosa[1–3]. Activated *h*EP can stimulate the conversion of trypsinogen to trypsin via cleavage of a specific trypsinogen activation peptide, namely Asp-Asp-Asp-Asp-Lys (DDDDK), which is highly conserved in vertebrates[4–6]. Then trypsin initiates a cascade of activations of various pancreatic zymogens, for example, chymotrypsinogen, proelastase, procarboxypeptidase, and prolipase, to regulate nutrient absorption and metabolism[2,7–10]. *h*EP also plays a pivotal role in body homeostasis[3,11–14]. Duodenopancreatic reflux of activated *h*EP can trigger a pancreatic enzyme cascade leading to acute necrotizing pancreatitis[2,15–18], whereas translocation of *h*EP into the pancreatic-biliary tract is associated with pancreatitis[19], suggesting *h*EP as a potential drug target for pancreatitis treatment[10,20,21].

*h*EP is a type II transmembrane serine protease that is synthesized as a catalytically inactive zymogen that can be activated by trypsin or related proteases[6,22,23]. However, the mechanism of *h*EP activation

remains to be established[23,24]. *h*EP contains a multi-domain heavy chain and a catalytic light chain, linked by a disulfide bond[25]. The heavy chain contains a single-helix transmembrane (TM) domain and seven structural motifs: one copy of Sea urchin sperm protein, Enteropeptidase, and Agrin (SEA), meprin-like domain (MAM), and Scavenger Receptor Cysteine-rich Repeat (SRCR); and two copies of Low-Density Lipoprotein Receptor (LDLR) and Complement, Urchin embryonic growth factor, and Bone morphogenetic protein-1 (CUB)[25,26] (Fig. 1a). The functions of these domains and motifs may be involved in protein anchoring, macromolecular substrate recognition and inhibitor specificity[23,25,27,28]. The light chain is homologous to trypsin-like serine proteinases with a typical Asp-His-Ser (D-H-S) catalytic triad responsible for peptidase activity[10,27].

Structural exploration of full-length *h*EP is crucial for understanding how the heavy chain modulates the full function of *h*EP. Previous crystal structures of the EP light chain revealed that the light chain has a typical trypsin-like serine protease fold[10,29–31]. Structural

[1]Department of Gastroenterology, Changhai Hospital, Navy/Second Military Medical University, Shanghai, China. [2]Shanghai YueXin Life-Science Information Technology Co., Ltd, Shanghai, China. [3]These authors contributed equally: Xiaoli Yang, Zhanyu Ding, Lisi Peng. ✉e-mail: zhs_li@126.com; huanghaojie@smmu.edu.cn

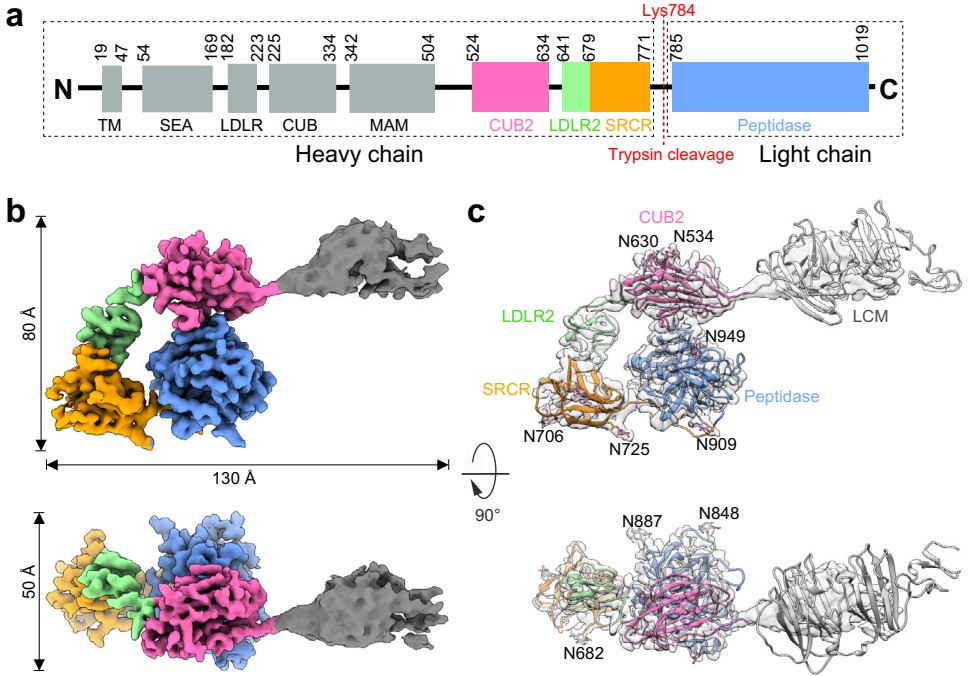

**Fig. 1 | Cryo-EM structure of *h*EP in the inactive state. a** Domains of *h*EP. Numbers indicate the N- or C-terminus positions of the domains in the amino acid sequence. The unexpressed domains (TM and SEA), as well as the flexible domains with a lower resolution (LDLR, CUB, and MAM), were colored in grey, CUB2 domain in pink, LDLR2 domain in light green, SRCR domain in orange, and the peptidase domain in dodger blue. This domain color scheme was used throughout this study. SEA: Sea urchin sperm protein, Enteropeptidase, and Agrin; MAM: meprin-like domain; SRCR: Scavenger Receptor Cysteine-rich Repeat; LDLR: Low-Density Lipoprotein Receptor; CUB: Complement, Urchin embryonic growth factor, and

Bone morphogenetic protein-1. **b** Cryo-EM reconstruction of inactive *h*EP. The reconstruction is a composite map generated from the locally refined *h*EP-core and the low-resolution region of *h*EP-complete. The domain color scheme follows that in (**a**). Unless otherwise stated, the same domain color scheme is applied to all figures. **c** Structural model of the inactive *h*EP, with the map displayed as semi-transparent. The nine detected N-linked glycans are shown as stick atomic models, and the amino acid residues to which they are covalently attached are labeled. LCM: LDLR, CUB, and MAM domains.

analysis of the EP light chain in complex with the small-molecule inhibitor camostat or a peptide substrate elucidated the mechanism by which EP was inhibited, as well as the mechanism of EP's proteolytic activity on trypsinogen[10,29]. Deletion of the EP heavy chain has been previously described to greatly affect inhibitor specificity and decrease proteolytic activity[23], but structural information about this heavy chain remains lacking, leaving a huge gap in our understanding of the functional mechanisms of EP. Structural analysis of full-length *h*EP and its complex with the physiological substrate will be an important step in outlining the framework for detailed characterization of *h*EP function and lay the foundation for *h*EP-targeted drug design.

In this work, we determine cryo-EM structures of *h*EP in the inactive and active states, and complexes with trypsinogen and nafamostat, a promising drug candidate targeting *h*EP as a treatment for *h*EP redundant disease. Our results provide structural and mechanistic insights into full-length *h*EP catalytic function, which advances our understanding of *h*EP dynamics and substrate coordination that may facilitate targeted therapy of *h*EP-related diseases.

## Results

### Cryo-EM structure of *h*EP in the inactive state

We initially determined the cryo-EM structure of the *h*EP protein starting from the SEA domain (residues 48–1019), i.e., with the TM domain truncated (Fig. 1a). However, this cryo-EM reconstruction was only obtained at low resolution (Supplementary Fig. 2c). The poor resolution was attributed to the presence of the SEA domain as previous reports indicated that the SEA domain might be involved in protein autocleavage, leading to increased instability[32,33]. Since the SEA domain has been shown to be dispensable for the enzymatic activity[23], a further truncated *h*EP including only residues 182–1019, i.e., also

leaving out the SEA domain (Fig. 1a), was used for further high-resolution structural analysis. The resulting purified *h*EP was a heavily glycosylated single-chain zymogen with a molecular weight (>130 kDa) (Supplementary Fig. 1a) higher than that theoretically predicted of the amino acids alone (95 kDa)[23]. An enzymatic activity assay performed in vitro confirmed its inactive state with no detectable substrate cleavage as previously described[2,23], and the *h*EP became active upon cleavage by trypsin into two chains (Supplementary Fig. 1a, b). Cryo-EM single-particle analysis of this inactive *h*EP yielded a cryo-EM reconstruction at 3.8 Å resolution, with the resolution of the core region reaching 2.7 Å (Fig. 1b, Supplementary Fig. 1c–g, and Supplementary Table 1), allowing explicit visualization of side-chain density of most residues except for those of surface loops L1, L2, and LD in the peptidase domain (Supplementary Fig. 2a, b).

The overall architecture of this inactive *h*EP was the same as that containing the SEA domain (residues 48–1019), showing a clamp shape adopted by the heavy chain with the light chain in the center (Fig. 1b and Supplementary Fig. 2c). The LDLR, CUB, and MAM (LCM) domains were observed to protrude from the remainder of the *h*EP-core region, which containing the CUB2, LDLR2, and SRCR domains of the heavy chain and the peptidase domain of the light chain, resulting in lower local resolution likely due to high flexibility of the LCM relative to the core region (Fig. 1c and Supplementary Fig. 1d, g). In addition to the disulfide bond (Cys772-Cys896) between the light and heavy chains, the CUB2 and SRCR domains stabilized the peptidase domain by forming hydrogen bonds between Glu581 and Arg871, and between Ser771 and Gln893 (Supplementary Fig. 2d). The CUB2 and SRCR domains were interconnected by the LDLR2 domain, and clamped the peptidase domain with a total interaction surface of 1163.2 Å² (Supplementary Fig. 2d).

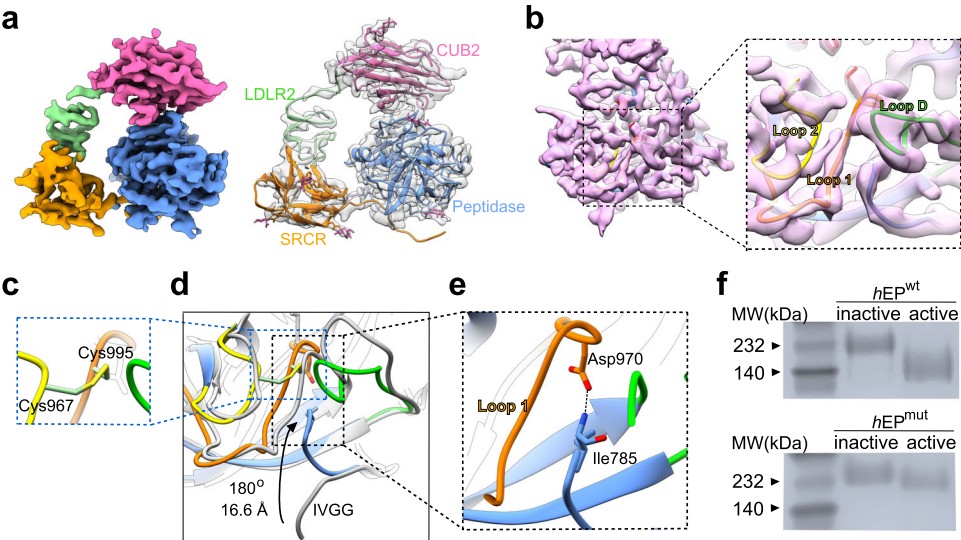

**Fig. 2 | Mechanism of the activation of *h*EP. a** Cryo-EM reconstruction of the active *h*EP^wt-core, with a structural model fitted in. **b** The peptidase domain portion of this reconstruction fitted with the corresponding model, with the inset showing a magnified view of the portion of this domain containing surface loops L1, L2, and LD. **c** Depiction of the inter-domain disulfide pair Cys967-Cys995, a feature forming the wall of the catalytic pocket. **d** Comparison of the structures of the catalytic pockets of active (with different colors) and inactive (grey) states. The black arrow indicates the rotation and shift of the IVGG sequence. **e** Depiction of the loop 1-stabilizing salt bridge between Asp970 and the newly exposed N-terminal amino group of Ile785. **f** Gel-shift assays for monitoring activation of *h*EP. The *h*EP^wt sample showed a large shift upon activation, while the shift was reduced for the *h*EP^mut. The images represent reproducible results in >3 independent experiments. Source data are provided as a Source Data file.

*h*EP is physiologically a highly glycosylated protein, containing 18 potential N-linked glycosylation sites, with nine of them located in the core region[23,34] (Supplementary Fig. 2e). Removal of its glycosylation has been shown to have an impact on its tertiary structure and enzymatic activity[25]. Additional density for glycans was observed at all nine N-glycosylation sites in the inactive *h*EP-core region (Fig. 1c and Supplementary Fig. 2f), suggesting a near-physiological state of our inactive *h*EP structure.

## Mechanism of the activation of *h*EP

Upon cleavage of the *h*EP into heavy and light chains by trypsin, the enzyme became fully active as shown by the protease activity assay (Supplementary Fig. 1a, b). The protruding LCM region was not resolved in the active *h*EP structure even after extensive 3D classifications (Supplementary Fig. 3c, f). There was also no evidence of cleavage of the LCM domain as we observed only two bands, corresponding to the intact heavy and light chains, upon activation (Supplementary Fig. 1a). This set of results indicated increasing dynamics of LCM relative to the *h*EP-core upon activation. Inspection of the cryo-EM structure of this active *h*EP-core at 3.2 Å resolution revealed an overall shape equivalent to that of the inactive state (Fig. 2a, Supplementary Figs. 3, 4a, and Supplementary Table 1). Superposition of the *h*EP-core regions of the active and inactive states revealed additional density near the catalytic site in the active state, corresponding to the surface loops L1, L2 and LD not resolved in the inactive state (Fig. 2b, Supplementary Figs. 2b, 4a). The catalytic pocket organization—involving three β-strands connected by surface loops L1 and L2, with the loops connected by the disulfide bond Cys967-Cys995[35] (Fig. 2c, d)—was indicated from our structures to be stabilized by the conserved peptidase sequence IVGG newly located at the N-terminus upon activation[36] (Fig. 2d). The IVGG sequence was observed to be flipped by ~180 degrees, with Ile785 shifted by ~16.6 Å (Fig. 2d). The newly exposed N-terminal amino group of Ile785 formed a salt bridge with the side chain of Asp970 to stabilize loop L1 (Fig. 2e). The light chain in our cryo-EM structure of the active *h*EP-core was quite similar to the published crystal structure of a *h*EP light chain variant (PDB ID: 4DGJ), with an

r.m.s. the deviation between them of only 0.7 Å (Supplementary Fig. 4b).

Upon activation, the conformation of the catalytic triad Asp876-His825-Ser971 appeared to have remained unchanged (Supplementary Fig. 4c). We mutated all three residues to Ala (namely *h*EP^mut) and observed the abolished proteolytic activity even in the active state (Supplementary Fig. 1b). Notably, native gel analysis showed a reduced shift of the mutated *h*EP core (*h*EP^mut-core) compared to the wild-type *h*EP core (*h*EP^wt-core) upon activation (Fig. 2f), likely caused by a change in charge. To investigate the influence of charge on the catalytic pocket arrangement, we determined the active *h*EP^mut-core structure at a resolution of 3.7 Å (Supplementary Figs. 3d, e, g–i, 4d, and Supplementary Table 1). The overall architectures and the catalytic triads of active *h*EP^mut, and *h*EP^wt cores were quite similar, with an r.m.s. deviation of 0.799 Å (Supplementary Fig. 4e–h), indicating that mutation of the catalytic triad abolished the catalytic activity without changing the catalytic pocket. In conclusion, our study revealed an activation mechanism of *h*EP involving a flipping of the IVGG loop to stabilize the surface loops around the catalytic sites; our results also suggested the surface charge provided by the catalytic triad to have no effect on *h*EP activation, but it may affect the catalytic substrate reaction by reducing substrate binding affinity.

## Inhibition of *h*EP activity

Inhibition of serine proteases has emerged as a novel therapy for many diseases, including Alzheimer's disease[37], autoimmunity disease[38], and COVID-19[39,40]. The development of specific inhibitors of *h*EP has aided clinical intervention in pancreatitis[10], with nafamostat and camostat as the most promising of these inhibitors (Fig. 3a, b). Nafamostat, whose structure is shown in Fig. 3c, is a broad-spectrum synthetic serine protease inhibitor, with an IC$_{50}$ value of 1.5 μM for the inhibition of *h*EP activity, and hence more potent than camostat in inhibiting other serine proteases[41,42]; moreover, nafamostat has been reported to be clinically useful in the treatment of acute pancreatitis[43,44].

We measured the binding affinity of nafamostat to active *h*EP to be 95.1 μM (Supplementary Fig. 5a). And active *h*EP incubated with nafamostat was subjected to cryo-EM structure determination

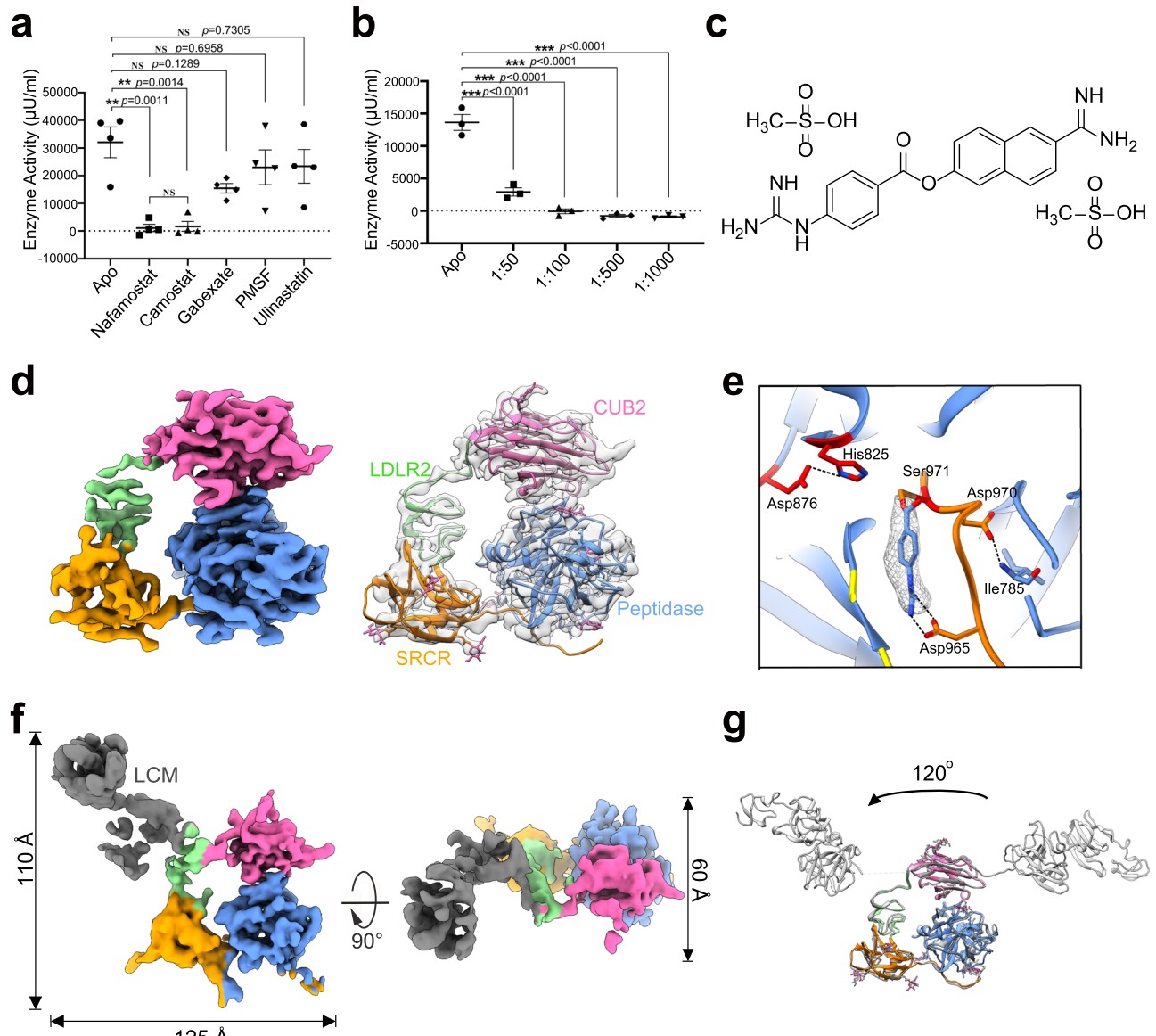

**Fig. 3 | Structural features of inhibited _h_EP. a** In vitro activity experiments under different inhibitor conditions for commercial _h_EP. Various compounds, including nafamostat, camostat, gabexate, PMSF, and ulinastatin, were tested to find the best inhibitor of _h_EP. _h_EP only (Apo) was also tested as a positive control. Data are presented as mean values ± SDs (standard deviations) from four biologically independent replicates. **b** In vitro _h_EP activity experiments. Commercial _h_EP samples were incubated with different amounts of nafamostat. _h_EP only (Apo) was also tested as a positive control. Data are presented as mean values +/- SDs from three independent experiments. In **a, b** significance was tested using one-way ANOVA.

**Significant probability level at _P_ < 0.01. ***Significant probability level at _P_ < 0.001. NS: Not Significant. Each _P_ value was adjusted to account for multiple comparisons. Source data are provided as a Source Data file. **c** Chemical structure of nafamostat mesylate. **d** Cryo-EM reconstruction of inhibited _h_EP-core, with a model fitted in. **e** Interactions between nafamostat and _h_EP. The density for nafamostat was shown. **f** Cryo-EM reconstruction of inhibited _h_EP-complete. **g** Overlay of the structural model of inactive _h_EP (all in grey) wit_h_ that of inhibited _h_EP-complete (with different colors), illustrating the change in the position of the LCM domain relative to the rest of the structure.

(Supplementary Fig. 5b–g). After 3D classification, ~58% of the particles showed no density for the LCM domain, resulting in a 3.1-Å-resolution map of the inhibited _h_EP-core (Fig. 3d, Supplementary Fig. 5e–g, and Supplementary Table 1). Although this inhibited _h_EP-core map was almost identical to the active map, with an r.m.s. deviation of 0.605 Å (Supplementary Fig. 6a, b), extra density was found in the catalytic pocket, which could be modelled as the reaction product of nafamostat with _h_EP. The model of this reaction product was found to covalently bind to the catalytic residue Ser971 and form an electrostatic interaction with the conserved Asp965 (Fig. 3e and Supplementary Fig. 6c), similar to the binding of camostat to the _h_EP light chain[10].

Cryo-EM reconstruction of another ~29% of the particles of the active _h_EP-nafamostat complex showed an intact _h_EP with the LCM

domain rotated ~120 degrees from the position of the inactive _h_EP LCM around the loop between MAM and CUB2 domains, with this inhibited _h_EP LCM in close proximity to the LDLR2 domain (Fig. 3f, g, Supplementary Figs. 5e, 6d, e). This considerable flexibility was thought to likely facilitate substrate recruitment and expose the catalytic site for subsequent cleavage. The light chain was observed to remain firmly clamped by the CUB2, LDLR2, and SRCR domains of the heavy chain within the core region (Fig. 3f), allowing for the intimate interaction between the heavy and light chains after the _h_EP-activating cleavage. The inhibitor nafamostat was thus found to covalently bind to Ser971 and induce an apparent change in the position of the _h_EP core region relative to the LCM domain.

### Substrate-engaged *h*EP

Given the critical role of pancreatic enzymes in the pathogenesis of acute pancreatitis, knowledge of the normal activation process of *human* zymogens initiated by *h*EP-cleaved trypsinogen to trypsin is apparently of great interest. In a previous study, the heavy chain of *h*EP was found to be necessary for efficient substrate recognition[23]. In order to capture the substrate-bound state of *h*EP, we incubated active *h*EP<sup>wt</sup> and *h*EP<sup>mut</sup> with the physiological substrate trypsinogen. As expected, active *h*EP<sup>wt</sup> cleaved the substrate, while the active *h*EP<sup>mut</sup> failed to do so (Supplementary Fig. 7a). The association of trypsinogen with active *h*EP<sup>mut</sup> was validated by a distinct migration pattern in native gel electrophoresis (Supplementary Fig. 7b). Thus, we incubated the active *h*EP<sup>mut</sup> with trypsinogen and purified the resulting complex using size exclusion chromatography, and did so in the presence of the cross-linker glutaraldehyde to maintain the complex for cryo-EM analysis (Supplementary Fig. 7c, d).

3D reconstruction of trypsinogen-bound *h*EP<sup>mut</sup> yielded a cryo-EM map at a resolution of 4.9 Å, with this relatively low resolution due to the particles having adopted a preferred orientation, resulting in a conformation of active *h*EP<sup>mut</sup> with fewer high-resolution features (Fig. 4a, Supplementary Fig. 8, and Supplementary Table 1). However, the active *h*EP<sup>mut</sup> alone can generate a normal *h*EP map, showing an overall shape equivalent to that of the active *h*EP<sup>wt</sup> with a well-organized catalytic pocket (Supplementary Fig. 4d–h). Thus, the engagement of substrate changed the performance of the active *h*EP<sup>mut</sup> under the cryoEM condition.

The overall architecture of the substrate-bound *h*EP<sup>mut</sup> was observed to be similar to that of the inactive *h*EP<sup>wt</sup>, with extra density observed to be attached to CUB2; this extra density could be attributed to trypsinogen (Fig. 4a, b and Supplementary Fig. 9a). A full-length model of trypsinogen including its uncleaved N-terminal peptide was constructed based on a prediction using AlphaFold2[45,46], and this model was docked as a rigid body into the extra density (Supplementary Fig. 9b). This apparent binding of trypsinogen to CUB2 validated the previous reports suggesting CUB2 to be essential for the recognition of natural substrates[47,48], and explaining the slow cleavage of trypsinogen by the isolated light chain[29]. Note, however, that the N-terminal activation peptide required a minor manual adjustment to be properly fit into the density (Fig. 4c, d and Supplementary

Fig. 9b, c). This N-terminal tail was inserted into the catalytic pocket, and the cleavage site was located at the center of the active site (Fig. 4d). This substrate-bound *h*EP structure provided valuable insights into the substrate recognition mechanism of CUB2. Nevertheless, please also note that the resolution of the resolved trypsinogen substrate, including its N-terminal tail, was not sufficient to assign individual amino acid residues. Detailed *h*EP-substrate interactions await higher-resolution structures.

## Discussion

Acute pancreatitis is a complex inflammatory disease of the pancreas, and its exact pathogenesis remains unclear[49,50]. *h*EP was discovered over 100 years ago as the physiological activator of trypsinogen[3,51,52]. However, the structural basis of full-length *h*EP processing, substrate binding, and small-molecule inhibition remains elusive. Inappropriate activation of trypsinogen, such as overactivation of trypsinogen by duodenopancreatic reflux of *h*EP, and premature or reduced degradation of trypsinogen caused by mutations in trypsinogen are thought to contribute to acute pancreatitis[16,18,53], and acquired *h*EP deficiency may be the origin of indigestion and malabsorption caused by functional pancreatic insufficiency[13,17,54]. Our structural study of *h*EP and its complexes has marked an important step in deciphering the pathogenesis of pancreatitis, and may open up a direction for the treatment of this disease.

Cryo-EM structures of *h*EP in inactive, active, inhibited and substrate-bound states revealed conformational changes of the *h*EP core region relative to the LCM domain, stabilization of the L1, L2, and LD loops around the catalytic site in the active structure, a mechanism of inhibition of *h*EP activity by nafamostat involving covalent binding, as well as a snapshot of substrate recognition. In general, *h*EP zymogen is anchored on the brush border of the duodenal and jejunal mucosa with a single transmembrane helix at the N-terminus, with this domain followed by the SEA, LCM, CUB2, LDLR2, SRCR and light-chain peptidase domains[1–3] (Fig. 1a). In the catalytic process, the zymogen first requires trypsin or other related proteases for activation[6,22,23] (Fig. 5). Upon activation of the zymogen, the flexible L1, L2 and LD loops form a complete and rigid catalytic pocket ready for performing catalysis. The binding of nafamostat or other covalent inhibitors to the catalytic site apparently results in a change of the conformation of the *h*EP core

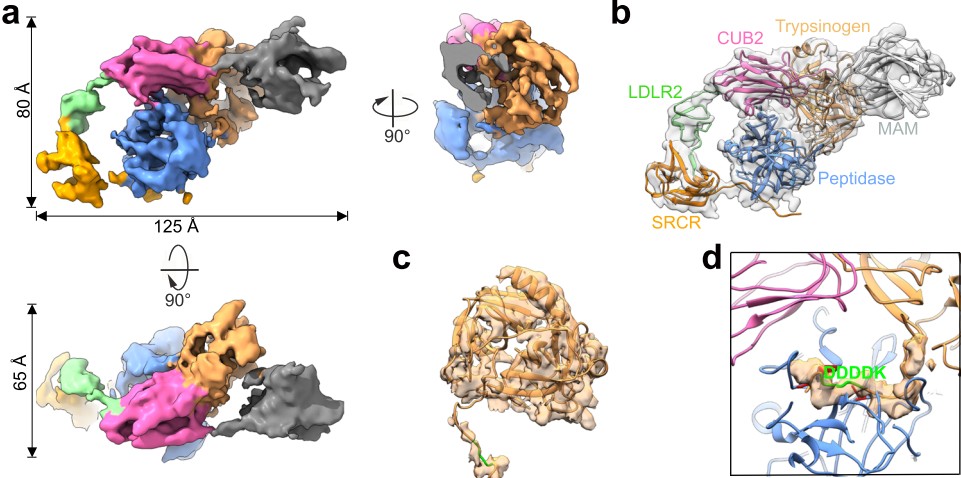

**Fig. 4 | Structural features of substrate-engaged *h*EP. a** Cryo-EM reconstruction of substrate-engaged *h*EP, with the extra density corresponding to trypsinogen colored sandy brown. **b** Structural model fit into the reconstruction of substrate-engaged *h*EP. The additional MAM and trypsinogen were fitted and labeled. **c** Flexible fitting of the AF2 predicted trypsinogen model into the subtracted extra density. The N-terminal tail with the cleavage site, DDDDK (colored green), was not

available in the published PDB structures, for they were all in the active state. Active *h*EP stimulates the conversion of trypsinogen to trypsin by cleaving this N-terminal tail. **d** Depiction showing the zoom-in view of the N-terminal tail of trypsinogen located in the catalytic triad of *h*EP. The catalytic triad on *h*EP is colored red. The cleavage site, DDDDK, is also labeled.

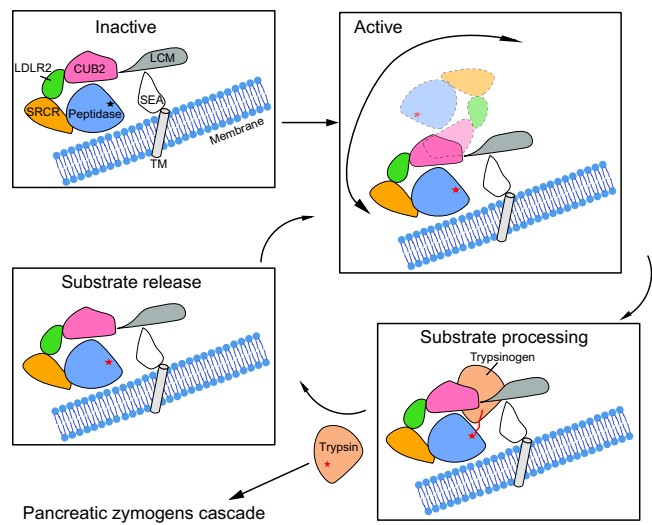

**Fig. 5 | Proposed working model of *h*EP.** In the catalytic process, the zymogen first requires trypsin or other related proteases for activation. Upon activation, the *h*EP-core becomes dynamic to facilitate the exposure of the catalytic site and recruitment of substrates. The complex marked by the dashed line displays one of the active states stabilized by the inhibitor. Substrates such as trypsinogen bind to CUB2 with the substrate N-terminal tail placed into the catalytic site for cleavage. Following cleavage of the N-terminal tail, trypsin is released to initiate the pancreatic zymogen cascade. *h*EP then returns to its dynamic state to recruit and catalyze the cleavage of more trypsinogen.

region. Substrates such as trypsinogen bind to CUB2 with the N-terminal tail placed into the catalytic site for cleavage (Fig. 5).

Our study has revealed the structural basis of *h*EP activation, substrate binding and nafamostat inhibition, and has thus provided an improved understanding of the *h*EP catalysis process and covalent inhibition mechanism that may shed light on *h*EP-associated pancreatitis.

## Methods
### Protein expression and purification
Two forms of wild type *h*EP (corresponding to residues 48–1019 and residues 182–1019) and a mutant *h*EP (H825A, D876A and S971A, residues 182–1019) were cloned using homologous recombination into, respectively, a pcDNA3 vector with an N-terminal signal peptide and 10 × His tags (HieffCloneTM One Step Cloning Kit, Yeasen). For each protein, HEK293F cells (A14635, ThermoFisher Scientific) were diluted to $3 \times 10^6$ cells/ml using serum-free SMM 293-TI medium (M293TI, Sino Biological Inc.), according to the protocols of manufacturers. Before the protein in each case was expressed, the cell was transfected by plasmid and Sinofection reagent (STF02, Sino Biological Inc.), with a ratio of transfection reagent: DNA is 5 μl: 1 μg. After 7 days of cell expansion, the cell supernatant was harvested, filtered, and loaded onto an Ni-NTA affinity purification column (GE Healthcare) in PBS buffer. Then the column was washed with a wash buffer made up of PBS buffer supplied with 20 mM imidazole, and the protein was eluted using 500 mM imidazole in PBS buffer. The eluted fractions containing *h*EP were identified using SDS-PAGE, pooled, and dialyzed at 4 °C against a buffer made up of 20 mM Tris-HCl, 20 mM NaCl, pH 7.6. The final pooled fractions were subjected to 8% SDS-PAGE and 8% native-PAGE in order to confirm the molecular weight and purity of the eluted *h*EP. For the native-PAGE, the samples were mixed with native dye and then subjected to the gel electrophoresis, which was run with a Tris-MOPS running buffer at 120 V and 4 °C for 1 h to separate the different conditions of *h*EP and then stained with 1% Coomassie Blue G250.

### Preparation of active *h*EP by using trypsin
Trypsin at 10 ng/μl (T1426, Sigma-Aldrich) (20 mM Tris-HCl, pH 4.0, 150 mM NaCl, and 5 mM CaCl₂) was incubated with purified 50 ng/μl *h*EP in buffer A (20 mM Tris-HCl, pH 7.6, 150 mM NaCl, and 5 mM CaCl₂) at a final mass ratio of 1: 100 for 2 h at 37 °C.

The *h*EP activity was determined using an enteropeptidase activity assay kit (K758-100, Biovision). Briefly, either a volume of 5 μl of *h*EP alone or that incubated with trypsin was combined with 95 μl of enteropeptidase assay buffer containing 2 μl of enteropeptidase substrate. The relative fluorescence units (RFU) (excitation/emission wavelengths = 380/500 nm) per well at 5 minutes and 0 minutes were subsequently measured using a fluorescence microplate reader (SpectraMax i3, Molecular Devices). Then the enteropeptidase activity levels of the samples were calculated from the fluorescence readings according to the manufacturer's instructions. Experiments were performed at least three times. GraphPad Prism software v8.3.0 was used to construct the statistic graphs. The proteins were also assessed using 8% SDS-PAGE and 8% native-PAGE.

The ability of *h*EP cleaves trypsinogen was determined by the trypsin activity colorimetric assay kit (MAK290, Sigma-Aldrich). Briefly, add 0, 2, 4, 6, 8, and 10 μl of p-NA Standard into a series of wells, respectively. Adjust volume to 50 μl/well with trypsin assay buffer to generate 0, 4, 8, 12, 16, and 20 nmol/well of the p-NA standard. A volume of 5 μl of *h*EP, active-*h*EP, *h*EP^mut, and active *h*EP^mut (50 ng/μl) in buffer A were combined, respectively, with 95 μl of trypsin assay buffer containing 2 μl of trypsin substrate and 5 μl trypsinogen (100 ng/μl) (T1143, Sigma-Aldrich, 20 mM Tris-HCl, pH 4.0, 150 mM NaCl, and 5 mM CaCl₂), except the trypsinogen control. The optical density (OD₄₀₅ ₙₘ) per well at 10 minutes and 0 minutes was subsequently measured spectrophotometrically on a microplate reader (SpectraMax i3, Molecular Devices). Then the trypsin activity levels of the samples were calculated from the optical density readings according to the manufacturer's instructions. Experiments were performed at least three times. GraphPad Prism software v8.3.0 was used to construct the statistic graphs.

### Preparation of inhibited *h*EP
Various compounds, including nafamostat, camostat, gabexate, PMSF, and ulinastatin (Topscience), were first tested to find the best inhibitor of *h*EP. PMSF was diluted in DMSO and the other above inhibitors were diluted in buffer A, in each case to a final concentration of 10 mg/ml for screening. Each inhibitor solution was then mixed with 5 μl of the solution of active *h*EP (50 ng/μl) at a molar ratio of 3000:1 at 37 °C for 30 minutes. Then the *h*EP enzymatic activity of each mixture was determined using the above-mentioned enteropeptidase activity assay kit (K758-100, Biovision). GraphPad Prism software v8.3.0 was used to construct the statistic graphs.

As stated in Results, nafamostat was more potent than camostat in inhibiting other serine proteases[41,42], and reported to be clinically useful in the treatment of acute pancreatitis[43,44], hence it was chosen to find the best inhibit efficacy. Samples of the nafamostat solution (10 mg/ml) were mixed with samples of 5 μl of active *h*EP (50 ng/μl) at molar ratios of 50:1, 100:1, 500:1, and 1000:1, respectively. The *h*EP activity of each mixture was then determined as mentioned above. GraphPad Prism software v8.3.0 was used to construct the statistic graphs.

### Preparation of substrate-bound *h*EP
A mixture containing a 1:5 molar ratio of *h*EP (1 mg/ml in buffer A) to trypsinogen (1 mg/ml in 20 mM Tris-HCl, pH 4.0, 150 mM NaCl, and 5 mM CaCl₂) was incubated in 1× PBS at 37 °C for 30 minutes, protected from light; then the products of this incubation were detected using native-PAGE to evaluate the association of trypsinogen with *h*EP. In addition, two experiments involving each an incubation of a mixture of

the *h*EP^mut (1 mg/ml in buffer A) with trypsinogen (1 mg/ml) at a 1:5 molar ratio were also conducted for 30 minutes at room temperature in the dark with one experiment also including 0.05% glutaraldehyde to crosslink the *h*EP^mut and trypsinogen in order to stabilize their complex, and the other experiment without crosslinker added; in the former experiment, the cross-linking was stopped after the 30 minutes of incubation by adding Tris, pH 8.0, to a final concentration of 1 mM. To obtain the highest purity of substrate-bound *h*EP^mut, gel filtration was performed on each incubated sample by passing the sample through a Superdex 200 (24 ml) column (GE Healthcare) with buffer A. The gel fractions containing trypsinogen-*h*EP^mut were identified by using 8% SDS-PAGE, and proteins were concentrated for cryo-EM data collection.

### Surface plasmon resonance analysis

Commercial *h*EP (7136-50, BioVision) was immobilized onto a CM5 sensor chip surface by using the NHS/EDC method with a Biacore 8k (Cytiva) and 1× PBS-T running buffer (1× PBS with 0.05% Tween-20). Then nafamostat solutions at concentrations of 1.95 μM, 3.9 μM, 7.8 μM, 15.625 μM, and 31.25 μM in the 1× PBS-T buffer were injected to flow over a different chip at a rate of 30 ml/min. The binding is monitored by subsequent changes in the refractive index of the medium close to the sensor surface upon injection, then the quantitative binding parameters were obtained. The resulting surface plasmon resonance data were analyzed using Affinity implemented in Biacore Insight Evaluation Software v3.0.

### Cryo-EM sample preparation and data collection

Samples with a volume each of 3 μl and concentrations of 1–1.5 mg/ml were placed onto glow-discharged holey amorphous nickel-titanium alloy film supported by 400-mesh gold grids[55], then blotted by deploying a Vitrobot Mark IV (FEI/Thermo Fisher Scientific), and flash frozen in liquid ethane.

Images were taken by using a Titan Krios transmission electron microscope (Thermo Fisher Scientific) operated at 300 kV and equipped with a Bio Quantum post-column energy filter with a zero-loss energy selection slit set to 20 eV. Images were collected by using a K2 Summit direct electron detector (Gatan) (K3 for the substrate-bound dataset) in super-resolution counting mode, corresponding to a pixel size of 0.523 Å (0.425 Å for the substrate-bound dataset) at the specimen level. Except for that in the substrate-bound dataset, each movie was dose-fractioned into 36 frames with a dose rate of 8 e per pixel per second on the detector. The exposure time was 7.2 s with 0.2 s for each frame, generating a total dose of ~52 e⁻/Å². Defocus values varied from −0.7 to −3.5 μm. All of the images were collected by using the SerialEM 3.9.0 automated data collection software package[56]. For the substrate-bound dataset, each movie was dose-fractioned into 40 frames with a dose rate of 16 e per pixel per second on the detector. The exposure time was 2.18 seconds, generating a total dose of ~50 e⁻/Å². Defocus values varied from −1.4 to −2.4 μm. All of the images were collected by using EPU 2.8.1 (Thermo Fisher Scientific).

### Cryo-EM data processing

Unless otherwise specified, single-particle analysis was mainly executed in cryoSPARC 2.15[57], including patch motion correction and patch CTF estimation. In the dataset collected from inactive *h*EP, the particles were auto-picked by using the blob picker in cryoSPARC[57], generating a dataset of 5,550,766 particles. After 2 rounds of 2D classification (particles binned 4 × 4 during extraction), the good class averages appeared white on a black background, showing internal secondary structural elements. The class averages showing relatively few features and a noisy background were discarded. The remaining 1,527,467 particles were subjected to ab initio reconstruction, followed by one round of heterogeneous refinement. After visual inspection of the resulting 4 classes using UCSF Chimera 1.14[58], the map in class one

showed clear features with continuous density and could be made to match the 2D class averages. Thus, 474,076 particles within class one were selected to perform another round of 2D classification in order to generate good class averages as the template for picking particles in the remaining datasets. Four rounds of heterogeneous refinement were performed to classify the particles from the template picker, resulting in one class with good features. The selected good classes from Blob and template pickers were combined together and further classified by carrying out two rounds of heterogeneous refinement. The particles from the good class were re-extracted with the original size (pixel size = 1.046 Å), and were subjected to further classification. The bad class was removed to obtain a cleaner particle stack, and the remaining three classes were combined to be reconstructed into a complete map, while the core map was locally refined by applying a *h*EP-core mask. The locally refined *h*EP-core and the low-resolution region in the whole map were merged in UCSF Chimera 1.14 and used for subsequent model building and analysis.

For all the other datasets, the selected reference and good particles from the dataset of inactive *h*EP constituted, respectively, the templates for particle picking and seed for seed-facilitated 3D classification[59]. The remaining procedures followed that used for the inactive dataset. Directional Fourier shell correlation (dFSC) curves were calculated as described[60], and the nominal resolution was estimated from the averaged FSC using the FSC = 0.143 criterion[61]. The local resolution was estimated either by using ResMap 1.1.4[62] or deepEMhancer 0.13[63].

### Model building and validation

The model of inactive *h*EP predicted by AlphaFold2[45,46] was initially fitted into the density maps using UCSF Chimera 1.14[58] and manually adjusted using COOT 0.8.9.1[64]. All the high-resolution models were subjected to multiple rounds of real-space refinement against the corresponding maps by using PHENIX 1.17.1[65].

In the inhibited *h*EP-complete structure, the EP-core from inhibited *h*EP-core structure and the LCM domain from inactive *h*EP-complete structure were fitted into the density map as rigid bodies. In the substrate-engaged *h*EP map, the EP model from inactive *h*EP-complete structure and a full-length model of trypsinogen predicted using AlphaFold2, were fitted into the density map as rigid bodies, except that the N-terminus of the substrate was manually adjusted. Thus, the related overall structure models were not further flexible refined.

The nafamostat and glycans were added using COOT 0.8.9.1[64]. To identify the glycans, we referred to the information in Uniprot database (https://www.uniprot.org/uniprot/P98073#ptm_processing). Uniprot showed 18 potential N-linked glycosylation sites, displaying GlcNAc-branched N-glycans. Nine N-linked GlcNAc were clearly detected in our cryo-EM maps of the *h*EP-core structure.

Model validation was performed following the phenix.molprobity protocol[65,66]. UCSF Chimera 1.14[58] and ChimeraX 1.1[67] were used for map segmentation and figure generation.

### Reporting summary

Further information on research design is available in the Nature Research Reporting Summary linked to this article.

## Data availability

The data that support this study are available from the corresponding authors upon reasonable request. EM maps have been deposited in the Electron Microscopy Data Bank (EMDB) under accession codes of EMD-32715 (Inactive-EP), EMD-32714 (Active-EP), EMD-32716 (Active-mutEP), EMD-32717 (Inhibited-EP), EMD-32828 (Inhibited EP-complete), and EMD-32829 (Substrate bound EP). Atomic coordinates have been deposited in the Protein Data Bank (PDB) under accession numbers of 7WQX (Inactive-EP), 7WQW (Active-EP), 7WQZ

(Active-mutEP), and 7WR7 (Inhibited-EP). For the two low resolution maps, inhibited *h*EP-complete and substrate-bound *h*EP, we deposited structural models only modeled as poly-Ala in the PDB under accession numbers of 8H3U (inhibitor-bound EP, polyA model) and 8H3S (sub-strate-bound EP, polyA model). Other structural model used in this study is available in the PDB with entry code of 4DGJ (X-ray model of *h*EP light chain variant). The source data underlying Figs. 2f, 3a, b, Supplementary Figs. 1a, b, 7 are provided as a Source Data file. Source data are provided with this paper.

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

## Acknowledgements

We thank members of the cryo-EM facility of Southern University of Science and Technology for providing facility support. We also thank Hong Wu and Zhangyun Jiang from Shanghai YueXin Life-Science Information Technology Co., Ltd for help with computation. This work was supported by National Natural Science Foundation of China Grants 82020108005 (to Z.S.L.) and 82022008 (to H.J.H.), and a grant from the technology support plan of China 2015BAI13B08 (to Z.S.L.).

## Author contributions

Conceived and designed the experiments: X.L.Y., Z.Y.D., L.S.P., and H.J.H. Designed constructs for structural studies: X.L.Y., L.S.P. and Q.Y.S. Protein purification: L.S.P., Q.Y.S., F.C., and D.Y.Z. Performed functional analysis: X.L.Y., Z.Y.D., C.C.X., K.L.L., H.Y., S.Y.L., Z.S.L., and H.J.H. Performed EM data collection and analysis: X.L.Y., and Z.Y.D. Structure reconstruction: X.L.Y., and Z.Y.D. Model building: Z.Y.D. SPR analysis: X.L.Y. and L.S.P. Wrote the paper: X.L.Y., Z.Y.D., L.S.P., and H.J.H.

## Competing interests

The authors declare no competing interests.
