## [Peer Review File · Nature Communications]

Cryo-EM structures reveal the activation and substrate recognition mechanism of human enteropeptidaseReviewers' Comments:

Reviewer #1:

Remarks to the Author:

In the manuscript entitled "Cryo-EM structures reveal the activation and substrate recognition mechanism of human enteropeptidase" by Yang et al. (Manuscript NCOMMS-22-10814) the authors present a number of different structures of human enteropeptidase (EP) as determined by cryo-electron microscopy.

The current version of the manuscript needs major improvement. Many parts of the manuscript are floppy written, lacking detail information on the obtained results. This is also reflected in the Materials and Methods section, which provides just minimal information for example on protein production for structural analysis.

- Do the authors obviously present a structure of substrate-bound EP, but the substrate is quite distinct of the active site. What could be a potential trigger to bring the substrate in close vicinity to the active site?
- Could the crosslinking approach have an influence on the mobility of certain domains?
- How were the inhibitors tested?
- Two more low resolution structures are mentioned in Table S1. What about these structures? Are they discussed within the manuscript?

- Page 3, second paragraph: write out the abbreviations of the domains (SEA, MAM,.....)
- Through out the manuscript replace "KDa" by "kDa"
- Page 6, last line: replace "crystallized model" by "crystal structure"
- Page 8: it is not surprising that a substitution of the catalytic residues by alanine residues diminishes activity and hence it is no surprise that the variant structure is basically identical compared to the wt enzyme.
- The authors are sometimes quite imprecise: "almost identical" please refer here to rmsd values
- Page 8, second paragraph: again, what are "good" particles?
- Page 8, second paragraph: why is it surprising to identify density for the inhibitor if it was added to the enzyme for structural characterization?
- Page 8, second paragraph: unit for affinity is missing
- Page 8, second paragraph: "this inhibited structure EP-core map is almost identical" again imprecise!
- Page 9, first paragraph: "...clamp-hold by a few domains..." Which domains?
- Page 9, second paragraph: which are the well-resolved structures? It is very difficult to compare structures of different resolution in terms of B-factors. A covalently bound inhibitor will always stabilize the protein.
- Page 10, first paragraph: which "activated mutant EP" was used?
- Avoid the term "pseudo-atomic" it is miss leading. These are medium resolution structures.
- Material and Methods: Carefully check, many buffer compositions are missing, no information on centrifugation speed, wrong units, spaces between values and units are missing. Figure 3, panel D: what is meant by "The detected 9 N-linked glycans were indicated by atomic view."
- Figure 4, panel C: Labels are difficult so see. Maybe enlarge panel.
- Figure 5: please label the schematic drawing, it is very difficult to understand the drawing without labels.
- Check figures for duplication. It seems that Fig. 2 panel C and Fig. S4 panel G are identical beside the setting of labels!?
- Fig. S7 giving the different resolution of the structure a comparison of B-factors is not possible. I would remove Fig. S7. The authors should carefully check which version of PHENIX realspace.refine was used. In the older versions, the temperature factor was refined as "grouped B-factor"!

Cryo-EM structures:

I miss some refinement statistics for the structures: Molprobity Score, clash score, Ramachandran Z-

score, Cblam, beta-deviation, average B-factor for each polypeptide chain and inhibitor. Model-map scores CC values for mask and volume, EM-ringer score.

Additionally, required figures:

- Page 9, paragraph 3: discussion of surface charges. Here, an electrostatic surface is required to understand and visualize the changes in charge distribution on the surface of the protein.
- Density around the nafomastat,

Reviewer #2:

Remarks to the Author:

This study uses cryo-EM to solve the structures of human enteropeptidase, in the full length at inactive, active, and inhibitor bound states. Even though this important enzyme has been studied extensively previously, the full-length structure is reported for the first time. In addition, this work shows that cryo-EM structures can be used to describe the dynamic process of enzyme catalysis, which is an emerging subject in cryo-EM research [<https://doi.org/10.1146/annurev-biophys-100121-075228>]. The proposed working model for the catalytic mechanism of hEP, as shown in Figure 5, appears to be reasonable. It goes beyond the usual mechanism of the active site chemistry, to include the dynamics of the multi-domains of the enzyme. The conclusions and claims are generally supported by the results, with the exception of the glycan analysis as indicated below. The methodology used in the protein structure analysis is sound, but there is room for some improvement.

Specific issues:

1. Based on Table S1, the resolutions of the four deposited structures are all greater than 3.0 Angstrom, except that the inactive-EP is listed as 2.7/3.8, with 2.7 from local refinement. Why wasn't local refinement performed with the other structures to reach higher resolutions? There is a great deal of difference between >3 and <3 angstrom resolutions.
2. It is nice to identify the glycans as shown in Fig. S2G. However, the identity of glycan should be indicated. In addition, I could not find how the identity of each glycan is identified. Is it assisted by mass spec as it is usually done?
3. There are many typos and grammar errors, particularly in the legend of Figure 5.

Reviewer #3:

Remarks to the Author:

Yang and coworkers have carried out a comprehensive cryo-EM study on human enteropeptidase (hEP), which explains the most important aspects of its functionality. This membrane anchored multidomain serine protease is a crucial activator of trypsinogen in the digestive system, in particular in the intestines. In addition, dysregulation of hEP can result in pancreatitis, a life threatening disease. Given this significance, the design of the study, the experimental approach and the results are excellent, in particular, since only four structures of the enteropeptidase serine protease domain (light chain) were deposited so far. The authors solved these structures of an hEP construct comprising most of the extracellular domains in inactive (or zymogen), active, inhibited and substrate bound states at moderate resolution. This series of structures allows for defining the essential states of the catalytic cycle, whereby the inhibitor nafamostat may serve as lead compound for pharmaceutical drugs against pancreatitis. Unfortunately, the authors present the study in a manner that is not acceptable for publication in Nature Communications. Firstly, the text contains numerous, which means more than 100, deviations from standard English. It is imperative that a native speaker or an English scientific editing service corrects the manuscript. Secondly, nearly all figures are far too small, most labels cannot be read easily, and some relevant information is only displayed in supplementary figures. Otherwise, the work supports the conclusions and claims, without any additional evidence needed. Methodology, data analysis and their interpretation is overall sound, according to the standards in the field. Seemingly, the manuscript provides enough detail for the work to be

reproduced. As no further experiments are required, a thorough rewriting and reworking of the figures, which I consider a major revision, may suffice for publishing this highly interesting study.

MINOR POINTS

Lines 25/26: "atomic 2.7 Å resolution" is a meaningless notion. Atomic resolution is below 2 Å.

L68: Delete the Greek letters alpha/beta, "typical trypsin-like fold" is the correct term.

L70: What does "inhibitor specificity ... depends on the heavy chain" mean? Explain.

L88: Throughout the manuscript "We etc." should be avoided.

L88/99-100: The construct does not contain the SEA domain, which is not mentioned at all. Does this deletion have consequences for the functionality? In the worst case, the model of the catalytic cycle is quite different from the one presented by the authors in Fig. 5. In addition, the missing SEA domain has to be addressed in the Discussion.

L93: What is "as-produced"?

L106: Delete unambiguously.

L112: To my knowledge, AlphaFold cannot easily predict the arrangement of the domains in a multidomain protein. If possible, eliminate this part, as all the single domains of hEP have been solved independently, which should facilitate their placement in the electron density.

L132: How was protease activity measured? It has to be added to the methods section.

L146: "good match". What is the RMSD?

L160: finishes?

L160: Why was a triple mutant generated? His825Ala alone would have abolished any protease activity.

L173: Correct to "of serine proteases".

L189: Replace surprisingly and delete unambiguous.

L191: How was the binding affinity determined? Does 9.51E-05 mean 10 to the power -5?

L195: What is blockage occupancy?

L247: It seems a bit absurd to employ AlphaFold2 predictions for trypsinogen, for which experimental PDBs are available.

L261: Inappropriate activation should be further explained. Is it premature, exceeding or something else?

L276: Do you mean rather stable or rigid, when writing "steady"?

L282: I prefer to specify the resolution as 2.7 Å instead of "high-resolution".

Methods in general: Numbers and units are given with and without spaces between them in an inconsistent manner.

L351: Why Tmprss15? It is just hEP.

L370: How are "good" templates defined?

Figures in general: They are all too small, labels are hardly readable. Enlarge all of them, except for Fig. 5.

L480: Fig. S1 A-D and F-G should be part of a regular figure.

L508: Fig. S3 H-I could be part of a regular figure.

L537: Fig. S5 E-F could be part of a regular figure.

L558: What does "surface property" mean? Fig. S7B does not contain relevant information.

L561: Fig. S8 H-I could be part of a regular figure.

Response to reviewer #1's comments:

Reviewer #1 (Remarks to the Author):

In the manuscript entitled "Cryo-EM structures reveal the activation and substrate recognition mechanism of human enteropeptidase" by Yang et al. (Manuscript NCOMMS-22-10814) the authors present a number of different structures of human enteropeptidase (EP) as determined by cryo-electron microscopy.

1-1. The current version of the manuscript needs major improvement. Many parts of the manuscript are floppy written, lacking detail information on the obtained results. This is also reflected in the Materials and Methods section, which provides just minimal information for example on protein production for structural analysis.

Response: We thank the reviewer for careful evaluation of our manuscript. We have consulted a colleague to rewrite the entire manuscript and added all required details in the manuscript including the Materials and Methods section.

1-2. Do the authors obviously present a structure of substrate-bound EP, but the substrate is quite distinct of the active site. What could be a potential trigger to bring the substrate in close vicinity to the active site?

Response: We thank the reviewer for this question. In the substrate-bound hEP structure, the substrate actually has the specific N-terminal activation peptide (Asp-Asp-Asp-Asp-Lys, DDDDK) located at the center of the active site, which need to be cleaved by EP to activate the activity of the substrate in human or bovine^{1,2,3}. We realized that this was not obvious in the previous manuscript and have updated Fig. 4c-d in the revised manuscript and as follow for clarification.

Fig. 4 **c** Flexible fitting of the AF2 predicted trypsinogen model in the subtracted extra density. **d** Depiction showing the zoom in view of the N-terminal tail of trypsinogen located in the catalytic triad of hEP. The catalytic triad on hEP is colored red. The cleavage site, DDDDK, is also labeled.

1-3. Could the crosslinking approach have an influence on the mobility of certain domains?

Response: We thank the reviewer for this critical question. Crosslinking as a strategy to stabilize complexes without changing their conformations and activity has been widely used for cryo-EM structure determination^{4,5,6,7}. It has also been shown that cross-linking induces rather subtle structural changes and the overall fold is preserved even at a higher cross-linker concentration⁶. Previous study has shown that covalent cross-linking of bovine EP light chain can retain approximately 90 % of the enzymatic activity⁸.

1-4. How were the inhibitors tested?

Response: We thank the reviewer for the question and have added the description of the corresponding experimental details. Briefly, we used an excess of inhibitors (10 µg/µl), incubated with the active hEP at a molar ratio of 3000:1 at 37 °C for 30 min. The enzyme activity was then measured by the enteropeptidase activity assay kit (K758-100, Biovision). Fluorescence values at 5 min and 0 min were selected to calculate the enzymatic activity.

1-5. Two more low resolution structures are mentioned in Table S1. What about these structures? Are they discussed within the manuscript?

Response: These two low resolution structures are the inhibited hEP-complete structure and the substrate-bound structure, respectively. They were discussed in the manuscript. Starting from line 174, the inhibited hEP-complete structure was reconstructed from the dataset in which the inhibitor was present. It was an intact hEP with the LCM domain rotated ~120 degrees from the position of inactive hEP LCM around the loop between MAM and CUB2 domains, with this inhibited hEP LCM in close proximity to the LDLR2 domain. This considerable flexibility was thought to likely facilitate the substrate recruitment and to expose the catalytic site for subsequent cleavage.

Starting from line 207, the substrate-bound hEP exhibited an extra substrate density attached to CUB2. This is the first time to visually capture the substrate-bound hEP structure, providing valuable insights into the substrate recognition mechanism of CUB2.

1-6. Page 3, second paragraph: write out the abbreviations of the domains (SEA, MAM,.....)

Response: We have revised the manuscript accordingly as follow, starting from line 46: sea urchin sperm protein, enteropeptidase, and agrin (SEA), meprin-like domain (MAM), scavenger receptor cysteine-rich repeat (SRCR), low density

lipoprotein receptor (LDLR), and complement, urchin embryonic growth factor, and bone morphogenetic protein-1 (CUB).

1-7. Through out the manuscript replace “kDa” by “kDa”

Response: We have revised accordingly.

1-8. Page 6, last line: replace “crystallized model” by “crystal structure”

Response: We have revised accordingly.

1-9. Page 8: it is not surprising that a substitution of the catalytic residues by alanine residues diminishes activity and hence it is no surprise that the variant structure is basically identical compared to the wt enzyme.

Response: We thank the reviewer for pointing this out. We completely agree with you. We apologize for having omitted the reasons for EP^{mut} reconstruction. In order to capture the substrate-bound state of hEP, we incubated active hEP^{wt} and hEP^{mut} with the physiological substrate trypsinogen. As expected, active hEP^{wt} cleaved the substrate, while the active hEP^{mut} failed to do so (Supplementary Fig. 7a). The association of trypsinogen with active hEP^{mut} was validated by a distinct migration pattern in a native gel electrophoresis (Supplementary Fig. 7b). Thus, we incubated the active hEP^{mut} with trypsinogen and purified the resulting complex using size exclusion chromatography, and did so in the presence of the cross-linker glutaraldehyde to maintain the complex for cryo-EM analysis (Supplementary Fig. 7c-d). 3D reconstruction of trypsinogen-bound hEP^{mut} yielded a cryo-EM map at a resolution of 4.9 Å, with this relatively low resolution due to the particles having adopted a preferred orientation, resulting in a conformation of active hEP^{mut} with fewer high-resolution features (Fig. 4a, Supplementary Fig. 8, and Supplementary Table 1). However, the active hEP^{mut} alone can generate a normal hEP map, showing an overall shape equivalent to that of the active hEP^{wt} with a well-organized catalytic pocket (Supplementary Fig. 4d-h). Thus, the engagement of substrate changed the performance of the active hEP^{mut} under the cryo-EM condition.

1-10. The authors are sometimes quite imprecise: “almost identical” please refer here to rmsd values

Response: We thank the reviewer for the comment. We have added the r.m.s. deviation values wherever is appropriate. Line 136, the sentence now is “...with an r.m.s. deviation between them of only 0.7 Å ...”, line 146, “...with an r.m.s. deviation of 0.799 Å ...”, and line 169, “...with an r.m.s. deviation of 0.605 Å...”.

1-11. Page 8, second paragraph: again, what are “good” particles?

Response: We agree with the reviewer that “good particles” is imprecise, we have removed this term when rewriting the manuscript.

1-12. Page 8, second paragraph: why is it surprising to identify density for the inhibitor if it was added to the enzyme for structural characterization?

Response: We agree with the reviewer and removed “surprisingly” when rewriting the manuscript.

1-13. Page 8, second paragraph: unit for affinity is missing

Response: We have added the unit accordingly. Line 163, the sentence now is “..the binding affinity of nafamostat to active hEP to be 95.1 μ M.”

1-14. Page 8, second paragraph: “this inhibited structure EP-core map is almost identical” again imprecise!

Response: We have removed such imprecise description when rewriting the manuscript.

1-15. Page 9, first paragraph: “...clamp-hold by a few domains....” Which domains?

Response: We have revised this sentence in line 181 to “The light chain was observed to remain firmly clamped by the CUB2, LDLR2, and SRCR domains of the heavy chain within the core region (Fig. 3f).”

1-16. Page 9, second paragraph: which are the well-resolved structures? It is very difficult to compare structures of different resolution in terms of B-factors. A covalently bound inhibitor will always stabilize the protein.

Response: We have removed “well-resolved” and B-factor comparisons in the revised manuscript, as also suggested in 1-24.

1-17. Page 10, first paragraph: which “activated mutant EP” was used?

Response: Thank you for pointing it out. Here the “activated mutant hEP” denoted the activation form of triple mutant hEP (H825A, D876A and S971A). In the revised manuscript, we have denoted this kind of EP as hEP^{mut}.

1-18. Avoid the term “pseudo-atomic” it is miss leading. These are medium resolution structures.

Response: We have removed “pseudo-atomic” throughout our manuscript.

1-19. Material and Methods: Carefully check, many buffer compositions are missing, no information on centrifugation speed, wrong units, spaces between values and units are missing.

Response: We thank the reviewer for the comment. We have carefully checked the Material and Methods section and filled all the missing details.

1-20. Figure 3, panel D: what is meant by “The detected 9 N-linked glycans were indicated by atomic view.”

Response: Thank you for pointing it out. We have resolved the nine N-glycosylation sites in our cryo-EM structures, and filled these N-linked glycans with atom models which can be seen in the Fig. 1c, as well as in Fig. 2a, 3d and Supplementary Fig. 4d. The covalently attached amino acids were labeled in Fig. 1c. To avoid confusion, we removed this sentence in Fig. 2a, 3d and Supplementary Fig. 4d.

1-21. Figure 4, panel C: Labels are difficult so see. Maybe enlarge panel.

Response: We have revised accordingly.

1-22. Figure 5: please label the schematic drawing, it is very difficult to understand the drawing without labels.

Response: We have included labels.

1-23. Check figures for duplication. It seems that Fig. 2 panel C and Fig. S4 panel G are identical beside the setting of labels!?

Response: Thank you for your careful inspection. Because the triple mutation of hEP (hEP^{mut}) did not change its spatial structure, the catalytic pocket of active hEP^{wt} and active hEP^{mut} was the same. Thus, the Fig. 2 panel C and Supplementary Fig. 4 panel G are almost identical, not duplicated.

1-24. Fig. S7 giving the different resolution of the structure a comparison of B-factors is not possible. I would remove Fig. S7.

Response: We have revised accordingly as also addressed in 1-16.

1-25. The authors should carefully check which version of PHENIX realspace.refine was used. In the older versions, the temperature factor was refined as “grouped B-factor”!

Response: We have added the PHENIX version number (1.17.1) in the Methods session in line 398.

1-26. Cryo-EM structures:

I miss some refinement statistics for the structures: Molprobity Score, clash score, Ramachandran Z-score, Cblam, beta-deviation, average B-factor for each polypeptide chain and inhibitor. Model-map scores CC values for mask and volume, EM-ringer score.

Response: We have now added all the requested statistics in the table. Please also see below.

	Inactive-hEP	Active hEP ^{wild} -core	Active hEP ^{mut} -core	Inhibited hEP-core	Inhibited hEP-complete	Substrate-bound
PDB	7WQX	7WQW	7WQZ	7WR7	N/A	N/A
EMDB	32715	32714	32716	32717	32828	32829
Data collection						
EM equipment	FEI Titan Krios	FEI Titan Krios	FEI Titan Krios	FEI Titan Krios	FEI Titan Krios	FEI Titan Krios
Voltage, kV	300	300	300	300	300	300
Detector	K2	K2	K2	K2	K2	K3
Pixel size, Å	0.523	0.523	0.523	0.523	0.523	0.425
Electron dose, e ⁻ /Å ²	52	52	52	52	52	50
Exposure time, s	7.2	7.2	7.2	7.2	7.2	2.18
Frames	36	36	36	36	36	40
Dedocus range, µm	1.0-3.5	0.7-3.4	0.9-3.5	1.0-3.5	1.0-3.5	1.4-2.4
Reconstruction						
Software	cryoSPARC	cryoSPARC	cryoSPARC	cryoSPARC	cryoSPARC	cryoSPARC
Raw micrographs	4,190	3,061	2,647	3,283	3,283	7,613
Final particles	511,658	307,754	156,344	343,205	119,676	251,202
Symmetry	C1	C1	C1	C1	C1	C1
Final resolution, Å	2.7/3.8	3.2	3.7	3.1	4.7	4.9
Map-sharpening B factor, Å ²	-87.7/-171.7	-99.7	-112.4	-110.9	-140.0	-304.7
Refinement						
Software	PHENIX	PHENIX	PHENIX	PHENIX	N/A	N/A
Rms deviations						
Bond length, Å	0.0020	0.0015	0.0017	0.0018	N/A	N/A
Bond angle, °	0.47	0.42	0.49	0.48	N/A	N/A
Ramachandran plot statistics, %						
Preferred	93.99	93.29	91.46	91.26	N/A	N/A
Allowed	6.01	6.71	8.54	8.74	N/A	N/A
Outlier	0.00	0.00	0.00	0.00	N/A	N/A
Ramachandran plot Z-score, RMSD						
Whole	-3.61 (0.30)	-4.3 (0.31)	-4.46 (0.30)	-4.52 (0.30)	N/A	N/A
Molprobrity score	2.20	2.41	2.27	2.38	N/A	N/A
Clash score	9.03	8.99	9.9	8.86	N/A	N/A
CaBLAM outliers, %	4.43	5.12	5.74	5.33	N/A	N/A
Cbeta outliers, %	0.00	0.00	0.00	0.00	N/A	N/A
Average B-factor for protein and inhibitor	70.15	118.69	136.2	98.86	N/A	N/A
Map CC (mask / volume)	0.64 / 0.64	0.75 / 0.74	0.74 / 0.74	0.72 / 0.72	N/A	N/A
EM-ringer score	2.93	2.23	1.55	2.80	N/A	N/A

1-27. Additionally, required figures:

Page 9, paragraph 3: discussion of surface charges. Here, an electrostatic surface is required to understand and visualize the changes in charge distribution on the surface of the protein.

Response: In the revised manuscript we removed the surface charge discussion because the difference was very little. A figure is included below for your reference.

Fig. R1 Surface charge distribution of hEP in different states and the zoom in views of each catalytic pocket region.

1-28. Density around the nafamostat,

Response: The density of nafamostat, as well as the density around nafamostat has already been shown in Fig. 3e and Supplementary Fig. 6c. For the convenience of the reviewer and editor, we also show it below.

Fig. R2 Density around nafamostat. (a) Interactions between nafamostat and hEP. The density for nafamostat was shown. (b) The model and map fitting around nafamostat.

Reviewer #2 (Remarks to the Author):

This study uses cryo-EM to solve the structures of human enteropeptidase, in the full length at inactive, active, and inhibitor bound states. Even though this important enzyme has been studied extensively previously, the full-length structure is reported for the first time. In addition, this work shows that cryo-EM structures can be used to describe the dynamic process of enzyme catalysis, which is an emerging subject in cryo-EM research [<https://doi.org/10.1146/annurev-biophys-100121-075228>]. The proposed working model for the catalytic mechanism of hEP, as shown in Figure 5, appears to be reasonable. It goes beyond the usual mechanism of the active site chemistry, to include the dynamics of the multi-domains of the enzyme. The conclusions and claims are generally supported by the results, with the exception of the glycan analysis as indicated below. The methodology used in the protein structure analysis is sound, but there is room for some improvement.

Response: We thank this reviewer for the positive feedback and constructive comments. We agree with this reviewer that this study demonstrates the capability of cryo-EM to study dynamic process of enzyme catalysis. We have included the abovementioned citation in the introduction part.

Specific issues:

2-1. Based on Table S1, the resolutions of the four deposited structures are all greater than 3.0 Angstrom, except that the inactive-EP is listed as 2.7/3.8, with 2.7 from local refinement. Why wasn't local refinement performed with the other structures to reach higher resolutions? There is a great deal of difference between >3 and <3 angstrom resolutions.

Response: Thanks to the reviewer for the constructive comments. In the inactive-EP structure we first resolved the complete structure at 3.8 Å, in which local resolution of the core region was < 3.4 Å, and local resolution of the LCM region was > 4.6 Å. So, we further performed a masked focus refinement in cryoSPARC and achieved 2.7 Å resolution for the core region in the absence of LCM. In active hEP^{WT} and hEP^{mut} samples only core regions were resolved so no focus refinement was performed. In inhibited hEP sample we obtained two different classes in 3D classification, one was the core region with nafamostat and another one had a lower resolution with the rearranged core region and the LCM domain, both of which could only be refined separately.

2-2. It is nice to identify the glycans as shown in Fig. S2G. However, the identity of glycan should be indicated. In addition, I could not find how the identity of each glycan is identified. Is it assisted by mass spec as it is usually done?

Response: Thank this reviewer for the question. For the identity of glycan, we referred to the information in Uniprot database (https://www.uniprot.org/uniprot/P98073#ptm_processing). Uniprot showed 18 potential N-linked glycosylation sites, displaying GlcNAc-branched N-glycans. Nine N-linked GlcNAc were clearly detected in our cryo-EM maps of the hEP-core structure. We have now described this in the Methods section for clarification.

2-3. There are many typos and grammar errors, particularly in the legend of Figure 5.

Response: Thank this reviewer for the comments and we rewrote the entire manuscript, and paid particular attention to Figure 5 legend. For the convenience of the reviewer and editor, we also show the legend of Figure 5 below.

Fig. 5 Proposed working model of hEP. In the catalytic process, the zymogen first requires trypsin or other related proteases for activation. Upon activation, the hEP-core becomes extremely dynamic to facilitate the exposure of the catalytic site and recruitment of substrates. The complex marked by the dashed line displays one of the active states stabilized by the inhibitor. Substrates such as trypsinogen bind to CUB2 with the substrate N-terminal tail placed into the catalytic site for cleavage. Following cleavage of the N-terminal tail, trypsin is released to initiate the pancreatic zymogen cascade. hEP then returns to its dynamic state to recruit and catalyze the cleavage of more trypsinogen.

Reviewer #3 (Remarks to the Author):

Yang and coworkers have carried out a comprehensive cryo-EM study on human enteropeptidase (hEP), which explains the most important aspects of its functionality. This membrane anchored multidomain serine protease is a crucial activator of

trypsinogen in the digestive system, in particular in the intestines. In addition, dysregulation of hEP can result in pancreatitis, a life threatening disease. Given this significance, the design of the study, the experimental approach and the results are excellent, in particular, since only four structures of the enteropeptidase serine protease domain (light chain) were deposited so far. The authors solved these structures of an hEP construct comprising most of the extracellular domains in inactive (or zymogen), active, inhibited and substrate bound states at moderate resolution. This series of structures allows for defining the essential states of the catalytic cycle, whereby the inhibitor nafamostat may serve as lead compound for pharmaceutical drugs against pancreatitis. Unfortunately, the authors present the study in a manner that is not acceptable for publication in Nature Communications. Firstly, the text contains numerous, which means more than 100, deviations from standard English. It is imperative that a native speaker or an English scientific editing service corrects the manuscript. Secondly, nearly all figures are far too small, most labels cannot be read easily, and some relevant information is only displayed in supplementary figures. Otherwise, the work supports the conclusions and claims, without any additional evidence needed. Methodology, data analysis and their interpretation is overall sound, according to the standards in the field. Seemingly, the manuscript provides enough detail for the work to be reproduced. As no further experiments are required, a thorough rewriting and reworking of the figures, which I consider a major revision, may suffice for publishing this highly interesting study.

Response: We thank this reviewer for acknowledging the significance of our work and providing constructive suggestions. We have now followed the reviewer's suggestions to rewrite the entire manuscript, to enlarge all figures.

MINOR POINTS

3-1. Lines 25/26: "atomic 2.7 Å resolution" is a meaningless notion. Atomic resolution is below 2 Å.

Response: We have removed "atomic".

3-2. L68: Delete the Greek letters alpha/beta, "typical trypsin-like fold" is the correct term.

Response: We have revised accordingly.

3-3. L70: What does "inhibitor specificity ... depends on the heavy chain" mean? Explain.

Response: Previous study has shown that the heavy chain significantly affects the inhibit specificity and selectivity of protease inhibitor ⁹. We rewrote the sentence in line 59 as follow for clarification: "Deletion of the hEP heavy chain has been

previously described to greatly affect inhibitor specificity and decrease proteolytic activity”.

3-4. L88: Throughout the manuscript “We etc.” should be avoided.

Response: We have revised accordingly.

3-5. L88/99-100: The construct does not contain the SEA domain, which is not mentioned at all. Does this deletion have consequences for the functionality? In the worst case, the model of the catalytic cycle is quite different from the one presented by the authors in Fig. 5. In addition, the missing SEA domain has to be addressed in the Discussion.

Response: We rewrote the manuscript and now explained the truncation of SEA domain in the beginning of the Results section. Previous study showed that hEP (residues 182-1,019) has similar protease activity compare to full length hEP⁹.

In this study, the structure of hEP including the SEA domain (residues 48-1,019) was also reconstructed by cryo-EM (Fig. R3). From the 2D and 3D classification, the overall shape of the hEP-core is the same as that of the structure without SEA domain (residues 182-1,019), both showing the light chain firmly clamped by the heavy chain. However, the presence of SEA domain led to cryo-EM reconstruction at low resolution, which was consistent with previous reports that the SEA domain might involve in protein autocleavage, leading to increased instability^{10,11}. In addition, the SEA domain had been shown to be dispensable for the enzymatic activity⁹. Therefore, the truncated hEP of 182-1,019 residues leaving out the SEA domain was used for further high-resolution structural analysis.

Fig. R3 The influence of SEA domain. (a-b) 2D classification showed similar conformations for hEP regardless of the presence or absence of the SEA domain (indicated by red ellipses). (c) Comparison of the reconstructed 3D maps.

3-6. L93: What is “as-produced“?

Response: This was removed in the revised manuscript.

3-7. L106: Delete unambiguously.

Response: Done.

3-8. L112: To my knowledge, AlphaFold cannot easily predict the arrangement of the domains in a multidomain protein. If possible, eliminate this part, as all the single

domains of hEP have been solved independently, which should facilitate their placement in the electron density.

Response: We have revised accordingly.

3-9. L132: How was protease activity measured? It has to be added to the methods section.

Response: We have added the details in the Method part. Briefly, trypsin and the purified hEP were co-incubated at a mass ratio of 1:100 for 2 h at 37 °C. The protease activity of the recombination hEP and the incubated products was measured by the enteropeptidase activity assay kit (K758-100, Biovision). The relative fluorescence units (RFU) (excitation/emission wavelengths = 380/500 nm) per well at 5 minutes and 0 minutes were selected for calculation of the enzymatic activity.”

3-10. L146: “good match“. What is the RMSD?

Response: We have added the r.m.s deviation value. Line 136 in the revised manuscript, now the sentence is “...with an r.m.s. deviation between them of only 0.7 Å ...”

3-11. L160: minishes?

Response: This was removed in the revised manuscript.

3-12. L160: Why was a triple mutant generated? His825Ala alone would have abolished any protease activity.

Response: Thank you very much for pointing this out. According to the substrate binding model of bovine EP, His and Ser in the catalytic triad will covalently bind to Lys at P1 position of the substrate¹², indicating that Ser is also engaged in the substrate catalysis. This is also reflected by our inhibited hEP-core structure, with nafamostat covalently binds to the catalytic residue Ser971. Even though His is indispensable for efficient catalysis, substitution by an alanine does not render the protease completely inactive^{13,14,15}. In order to ensure that the substrate is not cleaved, we need to ensure that hEP is completely inactive, so we mutated the catalytic triad involved in catalysis.

3-13. L173: Correct to “of serine proteases“.

Response: Done.

3-14. L189: Replace surprisingly and delete unambiguous.

Response: Done.

3-15. L191: How was the binding affinity determined? Does 9.51E-05 mean 10 to the power -5?

Response: Thank you very much for pointing it out. The binding affinity was determined by the Surface plasmon resonance technology (SPR), which we have revised to 95.1 μM .

3-16. L195: What is blockage occupancy?

Response: This was removed in the revised manuscript.

3-17. L247: It seems a bit absurd to employ AlphaFold2 predictions for trypsinogen, for which experimental PDBs are available.

Response: We are sorry that we didn't explain why we used the trypsinogen model predicted by AlphaFold2. It is true that trypsinogen has many experimental PDB structures, which are all in the active state (residues 24-247). This means that the specific trypsinogen activation peptide (**Asp-Asp-Asp-Asp-Lys, **DDDDK, residues 16-23) has been already removed. Active hEP stimulates the conversion of trypsinogen to trypsin via cleaving this peptide. Therefore, we need the trypsinogen model with this un-cleaved peptide to serve as the substrate for hEP. Now we have revised our manuscript accordingly.

3-18. L261: Inappropriate activation should be further explained. Is it premature, exceeding or something else?

Response: We have revised our manuscript accordingly. Line 230, now the sentence is "...such as overactivation of trypsinogen by duodenopancreatic reflux of hEP, and premature or reduced degradation of trypsinogen caused by mutations in trypsinogen...".

3-19. L276: Do you mean rather stable or rigid, when writing "steady"?

Response: In the revised manuscript, we have removed this kind of description.

3-20. L282: I prefer to specify the resolution as 2.7 Å instead of "high-resolution".

Response: We have revised our manuscript accordingly.

3-21. Methods in general: Numbers and units are given with and without spaces between them in an inconsistent manner.

Response: We have revised our manuscript accordingly.

3-22. L351: Why TMRSS15? It is just hEP.

Response: This was removed in the revised manuscript

3-23. L370: How are “good“ templates defined?

Response: Sorry for the brief description on the definition of good templates. We have revised our manuscript accordingly. Starting from line 364, “In the dataset collected from inactive hEP, the particles were auto-picked by using the blob picker in cryoSPARC¹⁶, generating a dataset of 5,550,766 particles. After 2 rounds of 2D classification (particles binned 4×4 during extraction), the good class averages appeared white on a black background, showing internal secondary structural elements. The class averages showing relatively few features and a noisy background were discarded. The remaining 1,527,467 particles were subjected to ab initio reconstruction, followed by one round of heterogeneous refinement. After visual inspection of the resulting 4 classes using UCSF Chimera¹⁷, the map in class 1 showed clear features with continuous density and could be made to match the 2D class averages. Thus, 474,076 particles within class 1 were selected to perform another round of 2D classification in order to generate good class averages as the template for picking of particles in the remaining datasets.”

3-24. Figures in general: They are all too small, labels are hardly readable. Enlarge all of them, except for Fig. 5.

Response: Done.

3-25. L480: Fig. S1 A-D and F-G should be part of a regular figure.

L508: Fig. S3 H-I could be part of a regular figure.

L537: Fig. S5 E-F could be part of a regular figure.

L561: Fig. S8 H-I could be part of a regular figure.

Response: We respect the reviewer’s opinion. If a regular figure here is referring to a figure in the main text, we think it’s not appropriate to show these FSC curves and local resolution maps as main figures. Instead, they are more appropriate to show in the supplementary figure as in the original submission.

3-26. L558: What does “surface property“ mean? Fig. S7B does not contain relevant information.

Response: In the revised manuscript we removed the related surface charge discussion because the difference was very little, which was also addressed in 1-27.

References:

1. Guy O, *et al.* Activation peptide of human trypsinogen 2. *FEBS Lett* **62**, 150-153 (1976).
2. Guy O, *et al.* Two human trypsinogens. Purification, molecular properties, and N-terminal sequences. *Biochemistry* **17**, 1669-1675 (1978).
3. Le Huerou I, *et al.* Isolation and nucleotide sequence of cDNA clone for bovine pancreatic anionic trypsinogen. Structural identity within the trypsin family. *Eur J Biochem* **193**, 767-773 (1990).
4. Ke Z, *et al.* Structures and distributions of SARS-CoV-2 spike proteins on intact virions. *Nature* **588**, 498-502 (2020).
5. Xie SC, *et al.* The structure of the PA28-20S proteasome complex from *Plasmodium falciparum* and implications for proteostasis. *Nat Microbiol*, (2019).
6. Rozbeský D, *et al.* Impact of Chemical Cross-Linking on Protein Structure and Function. *Analytical chemistry* **90**, 1104-1113 (2018).
7. Poepsel S, *et al.* Cryo-EM structures of PRC2 simultaneously engaged with two functionally distinct nucleosomes. *Nat Struct Mol Biol* **25**, 154-162 (2018).
8. Wang JH, *et al.* Site-specific, covalent immobilization of an engineered enterokinase onto magnetic nanoparticles through transglutaminase-catalyzed bioconjugation. *Colloids and surfaces B, Biointerfaces* **177**, 506-511 (2019).
9. Lu D, *et al.* Bovine proenteropeptidase is activated by trypsin, and the specificity of enteropeptidase depends on the heavy chain. *J Biol Chem* **272**, 31293-31300 (1997).
10. Levitin F, *et al.* The MUC1 SEA module is a self-cleaving domain. *J Biol Chem* **280**, 33374-33386 (2005).
11. Johansson DG, *et al.* SEA domain autoproteolysis accelerated by conformational strain: mechanistic aspects. *J Mol Biol* **377**, 1130-1143 (2008).
12. Lu D, *et al.* Crystal structure of enteropeptidase light chain complexed with an analog of the trypsinogen activation peptide. *J Mol Biol* **292**, 361-373

(1999).

13. Nagel F, *et al.* Structural and Biophysical Insights into SPINK1 Bound to Human Cationic Trypsin. *Int J Mol Sci* **23**, 3468 (2022).
14. Corey DR, *et al.* An investigation into the minimum requirements for peptide hydrolysis by mutation of the catalytic triad of trypsin. *Journal of the American Chemical Society* **114**, 1784–1790 (1992).
15. Corey DR, *et al.* Trypsin specificity increased through substrate-assisted catalysis. *Biochemistry* **34**, 11521–11527 (1995).
16. Punjani A, *et al.* cryoSPARC: algorithms for rapid unsupervised cryo-EM structure determination. *Nat Methods* **14**, 290–296 (2017).
17. Pettersen EF, *et al.* UCSF Chimera—a visualization system for exploratory research and analysis. *Journal of computational chemistry* **25**, 1605–1612 (2004).

Reviewers' Comments:

Reviewer #1:

Remarks to the Author:

overall, the manuscript has improved. But still there are many open or unresolved questions:

44-53 – several grammatical errors, sentence building wrong!

56 – several unnecessary citations. Added sentence not well integrated

57 – complexed structures?

77 – Figure1: colors are not depicted – ich kann nichts erkennen durch die Farbtöne

94 – Table S1: EM-ringer score very high, Molprobity Score very high, average B-factors are not depicted for each chain! Why local refinement for Inactive-hEP, but not other structures? Why two un-refined structures depicted? Model building probably very bad?

94 – Figure S2f: Structure depicted with Hydrogens. Hydrogens only with Res <1.7Å. Hydrogens have to be removed and the structure refined again. This would require an updated PDB submission.

133 – 16.6Å are stated to stabilize loop L1. Resolution is 3.0Å. The deviation of this resolution is higher than 3.6Å!

134 – Figure 2e: Saltbridge between I785 and D970. This does not make biochemically or physically sense!

150-153 – conclusion that abolishment of catalytic activity does not influence the folding of the catalytic pocket does not make sense. At least hydrogen bonds are abolished, this definitely changes the shape. Rmsd is stated for overall structure, not just catalytic pocket

170-174 – Nafomastat was "modeled" in the density. Does it make sense?

193 – Figure S7: SEC Chromatograms. I assume hEP in every fraction here according to SDS-PAGE. Not sure if this is validated data... mass spectrometry of bands needed.

206 – Figure S4g: again salt bridge between D and I. Res is 4.9Å. Not sure what to think of this... does absolutely not make sense

243 – Several domains are again depicted without further clarification. In my opinion a SEA or CUB2 domain is not general knowledge in structural biology.

245 – Figure5: is not well presented. Proposed mechanism. I am not sure if this can really be concluded from the structures...

Methods:

256-ongoing – the expression and purification part is very bad! No word about cultivation of cells oder origin of cells. No word about transfection ratios (This is extremely important!!!) no word about buffers (who can this be???). Just a dialysis buffer is mentioned... Hard to imagine that they used a PBS buffer for purification without protease inhibitors...

343 – however cryoEM preparation part is well explained!

395 – model building and validation again to briefly, which Coot versions were used? Which parts were masked during refinement? This also not fully clear after the data processing paragraph, but my also be just my impression.

1.1 – An additional author was introduced for rewriting. Where is a rewriting here? Just some paragraphs were added... additionally this author is missing in the author contributions

1.19 – not followed reviewers advice!

1.22 – also not followed

1.25 – also not followed. Probably they refined with grouped B-factor?

1.26 – still do not understand why the additional structures without refinement are depicted there (explanation in 1.5 is also miss-leading, did they compare the structures with unrefined data?)

1.26 – the Ramachandran Z-score is also bad, right?

Structures:

there are several major issues with the structures:

- 1.) Hydrogens should be not included
- 2.) Glycosilation makes absolutely no sence. Double check the chemistry how glcýcans are linked to protein. See also the validation repot of the PDB! Has the glycosilation chkecked by MS?
- 3.) Fig 3. panel c and panel e. I do not understand how the chemical formula of panle c should fit to panel e. I might have missed something, but to me the structure of nafamostat is substanstially different to the one fitted to the electron density volume

Overall the Ramachandran score for all structures is rather bad, indicating that some model rebuilding might be required.

Reviewer #2:

Remarks to the Author:

The authors have addressed all my concerns. The writing has been improved substantially.

We would like to thank the Reviewers for their evaluation of our work. We are very pleased that our revised manuscript has been substantially improved. and addressed all the concerns of Reviewer #2. As requested by Reviewer #1, we have provided detailed responses to each of the points below. Please note that our text is colored blue and the Reviewer's comments are in black.

Reviewer #1

Q1: overall, the manuscript has improved. But still there are many open or unresolved questions:

Response: First of all, thanks to the reviewer for agreeing that our revised manuscript has been improved. Then, we also carefully checked the entire manuscript and corrected spelling and grammar errors pointed out by the reviewer.

Q2: 44-53 – several grammatical errors, sentence building wrong!

Response: We have revised these sentences accordingly as follows: “The heavy chain contains a single-helical transmembrane (TM) domain and seven structural motifs: one copy of Sea urchin sperm protein, Enteropeptidase, and Agrin (SEA), meprin-like domain (MAM), and Scavenger Receptor Cysteine-rich Repet (SRCR); and two copies of Low-Density Lipoprotein Receptor (LDLR) and Complement, Urchin embryonic growth factor, and Bone morphogenetic protein-1 (CUB) (Fig. 1a). The functions of these domains and motifs may be involved in protein anchoring, macromolecular substrate recognition and inhibitor specificity. The light chain is homologous to trypsin-like serine proteinases with a typical Asp-His-Ser (D-H-S) catalytic triad responsible for peptidase activity.”

Q3: 56 – several unnecessary citations. Added sentence not well integrated

Response: We have revised the sentence accordingly as follow: “Previous crystal structures of the EP light chain revealed that the light chain has a typical trypsin-like serine protease fold^{10,29,30,31}.” In citation 10 entitled “Targeting enteropeptidase with reversible covalent inhibitors to achieve metabolic benefits”, the authors solved the crystal structure of the EP light chain inhibited by camostat. In citation 29, the authors presented “Crystal structure of enteropeptidase light chain complexed with an analog of the trypsinogen activation peptide” as the title stated. In citation 30, “Crystal structure of a supercharged variant of the human enteropeptidase light chain” was solved by the authors. In citation 31, “Structure basis for the unique specificity of

medaka enteropeptidase light chain” was revealed by the authors. All 4 citations are the crystal structures of the EP light chain, and the structures show that the light chain has a typical trypsin-like serine protease fold.

Therefore, we believe that these citations are necessary, and further support that our structure with heavy chain is important.

Q4: 57 – complexed structures?

Response: We have revised the sentence accordingly as follow: “Structural analysis of the EP light chain in complex with the small-molecule inhibitor camostat or a peptide substrate elucidated the mechanism by which EP was inhibited, as well as the mechanism of EP's proteolytic activity on trypsinogen.”

Q5: 77 – Figure1: colors are not depicted – ich kann nichts erkennen durch die Farbtöne

Response: Thank you for the constructive suggestion. We have added the colors as follow: “The unexpressed domains (TM and SEA), as well as the flexible domains with a lower resolution (LDLR, CUB and MAM) were colored in grey, CUB2 domain in pink, LDLR2 domain in light green, SRCR domain in orange, and the peptidase domain in dodger blue.”

Q6: 94 – Table S1: EM-ringer score very high, Molprobity Score very high, average B-factors are not depicted for each chain!

Response: EM-ringer and Molprobity scores of these cryoEM structures were actually low comparing to other structures, indicating structures presented in this manuscript are of high quality. Average B-factors are unnecessary for cryoEM structure validations. The PDB validation metrics also support that these structures are of high quality.

Q7: Why local refinement for Inactive-hEP, but not other structures?

Response: In the inactive-EP structure we first resolved the complete structure at 3.8 Å, in which the local resolution of the core region was < 3.4 Å, and local resolution of the LCM region was > 4.6 Å. So, we further performed a masked focus refinement in cryoSPARC and achieved 2.7 Å resolution for the core region in the absence of LCM. In active hEP^{WT} and hEP^{mut} samples only core regions were resolved, so no focus refinement was performed. In the inhibited hEP sample, we obtained two different classes in 3D classification: one was the core region with nafamostat and the other one had a lower resolution with the rearranged core region and the LCM domain, both of which could only be refined separately.

Q8: Why two un-refined structures depicted? Model building probably very bad?

Response: These two structures were in lower resolutions. In the inhibited hEP-complete structure, we fitted the EP-core from inhibited hEP-core structure and LCM domain from inactive hEP-complete structure into the density map using rigid body fitting. So as did to the substrate-engaged hEP map, except that the N-terminus of the

substrate was manually adjusted. Thus, the related overall structure models were not further refined. We have now added the description to the Methods session “Model building and validation”.

Although these two maps were in lower resolutions, they provided valuable insights into EP activation and substrate recognition mechanisms. Therefore, we included them in the manuscript.

Q9: 94 – Figure S2f: Structure depicted with Hydrogens. Hydrogens only with Res <1.7Å. Hydrogens have to be removed and the structure refined again. This would require an updated PDB submission.

Response: We should point out that in Fig. S2f, no Hydrogens were depicted. The chemical structure of GlcNAc-branched glycans was shown below, linked to Asn (N).

And the chemistry formula of glycan in the validation report of the PDB, as well as Fig. S2f are shown below. We don't think our structure needs to be re-refined or causes any confusion.

Q10: 133 – 16.6Å are stated to stabilize loop L1. Resolution is 3.xÅ. The deviation of this resolution is higher than X.6Å!

Response: We thank the reviewer for the comment. From the inactive to active structure, the shift of I785 in IVGG sequence was measured as 16.6 Å. To avoid confusion, we have revised the sentence. After the shift, the newly exposed N-terminal amino group of Ile785 formed a salt bridge with the side chain of Asp970 to stabilize loop L1, which was flexible in the inactive state (Supplementary Figs. 2b and 4a). This interaction was also observed in previous crystal structure of a hEP light chain variant (PDB ID: 4DGJ,

Supplementary Fig. 4b). We have updated Fig. 2e, Fig. S4g, and the related description in paragraph 1 on page 7 in the manuscript.

A similar interaction was revealed in transmembrane protease, serine 2 (TMPRSS2), and the post-activation Asp440:Ile256 salt bridge showed complete maturation of the protease (Fraser BJ, et al. Nat Chem Biol. 2022.). The related figure is shown below.

Q11: 134 – Figure 2e: Saltbridge between I785 and D970. This does not make biochemially or physically sense!

Response: Thank you for the comments. For more details, please refer to Q10.

Q12: 150-153 – conclusion that abolishment of catalytic activity does not influence the folding of the catalytic pocket does not make sense. At least hydrogen bonds are abolished, this definitely changes the shape. Rmsd is stated for overall structure, not just catalytic pocket

Response: Thank you very much for pointing it out. We compared the active hEP^{mut} and hEP^{wt} by overlaying their density maps, which showed good overlap (Supplementary Fig. 4e and 4h). The corresponding models also showed similar interactions (Supplementary Fig. 4g).

The overall shapes of these two structures were the same, at least at the main chain level. But limited to the resolution, the side chain interactions (hydrogen bonds) were not discussed, except for the large flipping of the IVGG loop to form the post-activation Asp970:Ile785 salt bridge showed complete maturation of the protease.

Q13: 170-174 – Nafomastat was “modeled” in the density. Does it make sense?

Response: Thank you for the comments. We have revised the sentence accordingly as follow: “... the catalytic pocket, which could be modelled as the reaction product of nafomastat with hEP. The model of this reaction product was found to ...”.

The figure above shows the chemical reaction of EP with camostat (Sun W, et al. J Pharmacol Exp Ther. 2020.). Though the chemical formula of nafomastat that we used to incubate with the active hEP was different from that of camostat (Fig. 3c), the acyl moiety of nafomastat was identical to that of camostat. This acyl moiety of nafomastat forms covalent bond with the serine residue in EP active site. It is in a similar inhibition mechanism with that of EP inhibited by camostat. Thus in Fig. 3e, the fitted molecule is the acyl moiety of nafomastat.

Sustained Inhibition of EP Leads to a Potential Therapeutic 513

e

Left: Crystal structure of the EP-compound 6 reaction product adduct, resolved to 2.19 Å (Sun W, et al. J Pharmacol Exp Ther. 2020.).

Right: the fitted molecule is the acyl moiety of nafomastat, which is covalently bonded with the nucleophile of EP active site serine residue.

Q14: 193 – Figure S7: SEC Chromatograms. I assume hEP in every fraction here according to SDS-PAGE. Not sure if this is validated data... mass spectrometry of bands needed.

Response: Thank you very much for the comments. We have confirmed that the purified protein is hEP by Cryo-EM structural reconstruction.

Q15: 206 – Figure S4g: again salt bridge between D and I. Res is 4.9Å. Not sure what to think of this... does absolutely not make sense

Response: First, I need to correct that this hEP^{mut} density map has a resolution of 3.7 Å, and the local resolution of the catalytic pocket is higher than 3.4 Å (Supplementary Fig. 3g and 3i). Then for more details about the salt bridge, please refer to Q10.

Q16: 243 – Several domains are again depicted without further clarification. In my opinion a SEA or CUB2 domain is not general knowledge in structural biology.

Response: We have clarified the full names of these abbreviated terms in the introduction. We have revised these sentences to make it clearer. Please refer to Q2.

Q17: 245 – Figure5: is not well presented. Proposed mechanism. I am not sure if this can really be concluded from the structures...

Response: In the proposed mechanism, in the inactive state, hEP zymogen is anchored on the brush border of the duodenal and jejunal mucosa with a single transmembrane helix at the N-terminus, with this domain followed by the SEA, LCM, CUB2, LDLR2, SRCR and light-chain peptidase domains. This can be concluded from the published literatures and our inactive hEP structure.

In the catalytic process, as previous study reports, the zymogen first requires trypsin or other related proteases for activation. Upon activation of the zymogen, the flexible L1, L2 and LD loops form a complete and rigid catalytic pocket ready for performing catalysis. This can be concluded from our active hEP structure.

Binding of nafamostat or other covalent inhibitors to the catalytic site apparently results in a change of the conformation of the hEP core region, as illustrated in the inset for active state.

Substrates such as trypsinogen bind to CUB2 with the N-terminal tail placed into the catalytic site for cleavage, which can be concluded from our substrate-bound hEP structure.

Methods:

Q18: 256-ongoing – the expression and purification part is very bad! No word about cultivation of cells oder origin of cells. No word about transfection ratios (This is extremely important!!!) no word about buffers (who can this be???).. Just a dialysis buffer is mentioned... Hard to imagine that they used a PBS buffer for purification without protease inhibitors...

Response: For the cell culture and transfection, we followed the product instructions for serum-free SMM 293-TI medium (M293TI, Sino Biological Inc.) and Sinofection reagent (STF02, Sino Biological Inc.) (<https://www.sinobiological.com/other->

products/sinofection-transfection-reagent-stf02), which have been already described in the Method part. The ratio of transfection reagent: DNA is 5 µl: 1 µg.

In fact, the protein was indeed purified in PBS buffer without any protease inhibitors.

Q19: 343 – however cryoEM preparation part is well explained!

Response: Thanks for the positive comments.

Q20: 395 – model building and validation again to briefly, which Coot versions were used? Which parts were masked during refinement? This also not fully clear after the data processing paragraph, but my also be just my impression.....

Response: Thank you very much for the comments. We have added the version numbers of COOT (0.8.9.1), as well as Chimera (1.14) and ChimeraX (1.1). Also all required details in the model building part have been included in the revised manuscript.

For the model refinement, no regions were masked.

Q21: 1.1 – An additional author was introduced for rewriting. Where is a rewriting here? Just some paragraphs were added... additionally this author is missing in the author contributions

Response: We have rewritten the entire manuscript with the help of our colleague, so no new authors were introduced.

Q22: 1.19 – not followed reviewers advice!

Response: We have double-checked the Materials and methods section and re-filled in all missing details.

Q23: 1.22 – also not followed

Response: We went through the figure and added all the labels that the shapes represented.

Q24: 1.25 – also not followed. Probably they refined with grouped B-factor?

Response: Actually, the model refinement was performed following the *phenix.real.space.refinement* protocol. We specified the operation that ran with global minimization and compute atomic displacement parameters (ADP or B-factors per-atom). Then the atomic models were colored according to the B-factor distribution in Chimera.

As requested by the reviewer before, we have removed the B-factor comparisons in the revised manuscript. Therefore in 1.25, we only included the PHENIX version number (1.17.1), without the details for B-factor.

Q25: 1.26 – still do not understand why the additional structures without refinement are depicted there (explanation in 1.5 is also miss-leading, did they compare the structures with unrefined data?)

Response: If this “refinement” here is referring to density map refinement, the answer is YES, and we have refined all the density maps in CryoSPARC.

If it is referring to model refinement, the answer is NO. For the detailed explanation, please refer to Q8.

Q26: 1.26 – the Ramachandran Z-score is also bad, right?

Response: As explained in Q6, these values are not typical standards for evaluating a cryoEM structure.

Structures:

there are several major issues with the structures:

Q27: 1.) Hydrogens should be not included

Response: As mentioned in Q9, no Hydrogens were depicted.

Q28: 2.) Glycosilation makes absolutely no sence. Double check the chemistry how glcýcans are linked to protein. See also the validation repot of the PDB! Has the glycosilation chchecked by MS?

Response: The way that glycans linked to protein is shown in Q9.

For the identity of glycan, we referred to the information in Uniprot database (https://www.uniprot.org/uniprot/P98073#ptm_processing). Uniprot showed 18 potential N-linked glycosylation sites, displaying GlcNAc-branched N-glycans. Nine N-linked GlcNAc were clearly detected in our cryo-EM maps of the hEP-core structure.

Q29: 3.) Fig 3. panel c and panel e. I do not understand how the chemical formula of panle c should fit to panel e. I might have missed something, but to me the structure of nafamostat is substanstially different to the one fitted to the electron density volume

Response: We thank the reviewer to point it out. Actually, Fig. 3 panel c shows the chemical formula of nafamostat that we used to incubate with the active hEP. In panel e, the fitted molecule is the acyl moiety of nafamostat, which is covalent bonded with the nucleophile of EP active site serine residue. It is in a similar inhibition mechanism with that of EP inhibited by camostat (Sun W, et al. J Pharmacol Exp Ther. 2020.). For more details, please refer to Q13.

Q30: Overall the Ramachandran score for all structures is rather bad, indicating that some model rebuilding might be required.

Response: The structures are of good quality, please also refer to Q6.

Reviewer #2

The authors have addressed all my concerns. The writing has been improved substantially.

Response: Thank the reviewer for the positive comment.

Reviewers' Comments:

Reviewer #1:

Remarks to the Author:

I am happy with the current version of the manuscript. The only minor point or suggestion I have to the authors in regard of the two low resolution structures. I have to admit that I just realized that the low resolution maps have been deposited in the EMDB. That is excellent. The authors might consider, that they can also deposit a structural model only modeled as poly-Ala or Calpha trace.

My previous statement that it is actually the reaction product of nafamostat, should be now clear to the general reader.

Just to be precise, the B-factors have been refined as "grouped B-factors" which is absolutely fine. In Supplementary Table S1, please change to "average grouped B-factor" and for the values remove the second decimal space.

REVIEWERS' COMMENTS

Reviewer #1 (Remarks to the Author):

I am happy with the current version of the manuscript. The only minor point or suggestion I have to the authors in regard of the two low resolution structures. I have to admit that I just realized that the low resolution maps have been deposited in the EMDB. That is excellent. The authors might consider, that they can also deposit a structural model only modeled as poly-Ala or Calpha trace.

Thank the reviewer for the positive comment. We have followed the suggestions that these two low resolution maps were deposited in PDB under accession code 8H3U and 8H3S.

My previous statement that it is actually the reaction product of nafamostat, should be now clear to the general reader.

Thanks for the positive comments.

Just to be precise, the B-factors have been refined as "grouped B-factors" which is absolutely fine. In Supplementary Table S1, please change to "average grouped B-factor" and for the values remove the second decimal space.

We have revised accordingly.